# PALLATOM-LIGAND: AN ALL-ATOM DIFFUSION MODEL FOR DESIGNING LIGAND-BINDING PROTEINS

**Haochen Wang**[1 3][*]**, Qianyi Wang**[2]**, Rui Ma**[3]**, Jiawei Guan**[3]**,
Weikun Wu**[3][†]**, Haobo Wang**[3][†]**, Jiayi Dou**[2][†]

[1]School of Information Science and Technology, ShanghaiTech University
[2]School of Life Science and Technology, ShanghaiTech University
[3]LEVINTHAL Biotechnology Co., Ltd.

## ABSTRACT

Small-molecule ligands extend protein functionality beyond natural amino acids, enabling sophisticated processes like catalysis, signal transduction, and light harvesting. However, designing proteins with high affinity and selectivity for arbitrary ligands remains a major challenge. We present Pallatom-Ligand, a diffusion model that performs end-to-end generation of ligand-binding proteins at atomic resolution. By directly learning the joint distribution of all atoms in the protein–ligand complexes, Pallatom-Ligand delivers state-of-the-art performance, achieving the highest *in silico* success rates in a comprehensive benchmark. In addition, Pallatom-Ligand's novel conditioning framework enables programmable control over global protein fold and atomic-level ligand solvent accessibility. With these capabilities, Pallatom-Ligand opens new opportunities for exploring the protein function space, advancing both generative modeling and computational protein engineering. The code is available at `https://github.com/levinthal/Pallatom-Ligand`.

## 1 INTRODUCTION

The ability to design proteins with high affinity and selectivity for a given small molecule is critical for the development of protein-based biosensors, therapeutics, and diagnostics. Traditional methods for engineering ligand-binding proteins for medical and biotechnological applications rely on laboratory-directed evolution, which introduces random mutations and selects the desired properties through multiple rounds of labor-intensive experiments (Boder et al., 2000). Computational protein design provides a complementary approach to generate *de novo* small molecule-binding proteins by directly modeling the protein-ligand interactions at atomic level (Tinberg et al., 2013; Dou et al., 2018; Polizzi & DeGrado, 2020). However, physics-based computational methods like Rosetta require expert-level biochemical intuition and are hampered by inaccurate energy evaluation (Dou et al., 2017).

Deep learning has revolutionized protein science, surpassing physics-based approaches in structure prediction, unconditional generation, and protein-protein interface design (Jumper et al., 2021; Watson et al., 2023; Abramson et al., 2024). Yet its application to designing small molecule-binding proteins remains limited. Current state-of-the-art models, such as RFdiffusionAA, CA_RFdiffusion, and RFdiffusion2, are insufficient in learning the atomic interactions at the protein-ligand interface and rely on a separate inverse-folding model to design the protein sequence. While these methods have achieved notable experimental successes, including the design of functional enzymes (Krishna et al., 2024; Lauko et al., 2025; Ahern et al., 2025), their experimental success rate remains low compared to ligand-free protein design and requires significant expert intervention.

We identify two fundamental challenges in applying deep learning methods to designing ligand-binding proteins: the lack of explicit atomic-level modeling of protein-ligand interactions, and the

---

[*]Work done during an internship at LEVINTHAL Biotechnology.
[†]Corresponding authors. Emails: Weikun Wu (weikun.wu@levinthal.bio), Haobo Wang (haobowang@levinthal.bio), Jiayi Dou (doujy@shanghaitech.edu.cn).

shortage of high-quality structural data. We hypothesize that an end-to-end approach learning the joint distribution of all atoms in a protein-ligand complex can improve both the modeling accuracy and the data efficiency of learning. This hypothesis is grounded in two observations. First, atoms are the fundamental unit of all molecules and provide a unifying representation across chemistry space. The remarkable success of AlphaFold3 in biomolecular structure prediction demonstrated the power of directly modeling individual atoms (Abramson et al., 2024). Second, the aforementioned backbone-only generative models lack the mechanism for atomic details in ligand and protein side chains to refine the backbone generation. This desired multi-level information exchange could be achieved through a unified architecture combining all-atom representation with an end-to-end training.

To test our hypothesis, we developed Pallatom-Ligand (Figure 1), a novel diffusion-based model that generates ligand-binding proteins with all-atom modeling. Our main contributions are summarized as below:

1. **A unifying representation for small molecules and unknown protein residues.** In Pallatom-Ligand, small molecules are directly represented by their atoms, while each amino acid residue is modeled as a generic molecule of 14 atoms. We implemented a novel ligand-aware all-atom diffusion transformer to enable global information exchange across the protein-ligand complex.

2. **Multi-level controls tailored for designing ligand-binding proteins.** We experimented with novel conditioning strategies tailored for designing ligand-binding proteins. Notably, we implemented global control over the protein tertiary structure to guide the model towards generating diverse folds; we also achieved atom-level control over the ligand solvent accessibility for downstream applications.

3. **Novel evaluation metrics to assess the model performance *in silico*.** We introduced AlphaFold3-based evaluation metrics specific for the task of designing ligand-binding protein. We performed a comprehensive benchmark study comparing RFdiffussionAA, RFdiffusion2, and Pallatom-Ligand. The results demonstrate that Pallatom-Ligand delivers state-of-the-art performance.

## 2 RELATED WORKS

### 2.1 ALL-ATOM PROTEIN DESIGN METHODS

Diffusion and flow-based generative models have demonstrated notable success in *de novo* protein design in the past years. Early generative models for protein design were focused on learning the geometric distribution of protein backbones. To list a few, FrameDiff and RFdiffusion operate on the SE(3) manifold using backbone local frames, while Chroma and Proteina work in Euclidean space to model backbone atoms (Yim et al., 2023; Watson et al., 2023; Ingraham et al., 2023; Geffner et al., 2025b). Concurrently, protein design methods based on those models follow a two-stage protocol where the protein backbones and sequences are generated separately, and evaluated post hoc by a structure prediction model such as Alphafold2 or ESMfold (Jumper et al., 2021; Lin et al., 2023).

More recently, efforts have shifted towards all-atom generative models that capture the joint distribution of structure and sequence. These approaches can be broadly categorized into three streams: **sequence-centric** methods, such as ProteinGenerator (Lisanza et al., 2023) and PLAID (Lu et al., 2024), which prioritize sequence generation or latent representations and rely on structural decoders to recover geometry; **structure-centric** models like Protpardelle (Chu et al., 2024) and Pallatom (Qu et al., 2024), which focus on geometric generation by either simultaneously updating side-chains or inferring sequences from denoised all-atom coordinates; and **unified** frameworks such as La-Proteina (Geffner et al., 2025a), which integrate both modalities into a shared latent space via VAEs to achieve coherent co-generation.

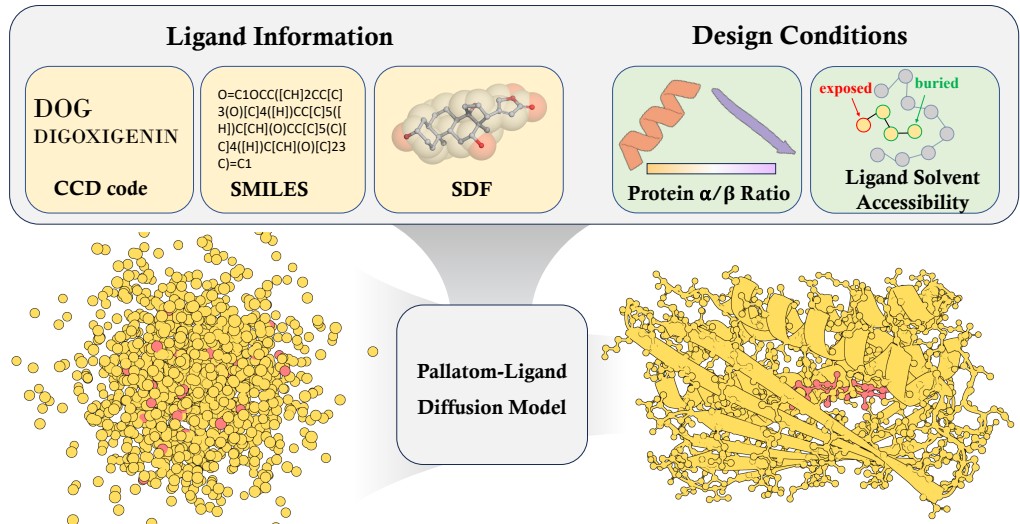

Figure 1: **Pallatom-Ligand Overview** Pallatom-Ligand generates an all-atom protein-ligand complex through an end-to-end diffusion process. The model takes two inputs: the chemical definition of the small-molecule ligand in one of the three formats (cream background; CCD: chemical composition distribution, SMILES: simplified molecular line entry system, SDF: structure data file) and the design conditions (mint background).

## 2.2 *De novo* DESIGN OF LIGAND-BINDING PROTEINS

*De novo* design of ligand-binding proteins is defined as the generation of novel proteins that can recognize a given small-molecule ligand. It was first demonstrated by knowledge-based methods such as RosettaMatch, RifDock, and van der Mer(vdM)-based methods (Tinberg et al., 2013; Dou et al., 2018; Polizzi & DeGrado, 2020). These methods start with pre-generated protein backbones called "scaffolds" and search for a combination of positions in each scaffold that can constellate a 3D pocket suitable for binding the target ligand. Their success rates depend on many factors, such as the choice of the scaffolds, the chemical complexity of the target ligand, and the evaluation of the binding site.

More recently, with the success of deep learning-based unconditional protein generation (Related Works 2.1 ), several generative models have been developed for ligand-binding proteins. RFdiffusionAA models protein backbones and small-molecule ligands as SE(3) frames and leverages the SE3-transformer to exchange information across the protein-ligand interface (Krishna et al., 2024). CA_RFdiffusion simplifies RFdiffusionAA backbone representation to $C_\alpha$ atoms only and directly operates on the atomic features to design ligand-aware protein backbones (Lauko et al., 2025). Along this line, RFdiffusion2 extends the original formulation by incorporating advanced conditions, such as unindexed motifs, ligand orientations, and relative solvent accessibility, to support enzyme design tasks (Ahern et al., 2025). Despite these advances, all three models remain backbone-only generators; the diffusion process does not involve protein side chains. Without explicit modeling of intricate atomic interactions at the protein-ligand interface, these models rely on user-provided constraints to obtain the desired spatial configuration. Such reliance on expert-crafted priors not only introduces strong biases in a case-dependent manner, but also limits their general usability.

## 3 METHOD

### 3.1 A UNIFYING SCHEME FOR ENCODING PROTEIN-LIGAND HYBRID SYSTEM

We adopt a unifying atom representation for encoding the chemical and structural information of protein-ligand hybrid system. For a small-ligand with $l$ atoms, its chemical complexity and structural connectivity are directly encoded at the atomic level. Each amino-acid residue in the protein chain is modeled as a generic chemical entity containing 14 atoms according to Pallatom's atom14

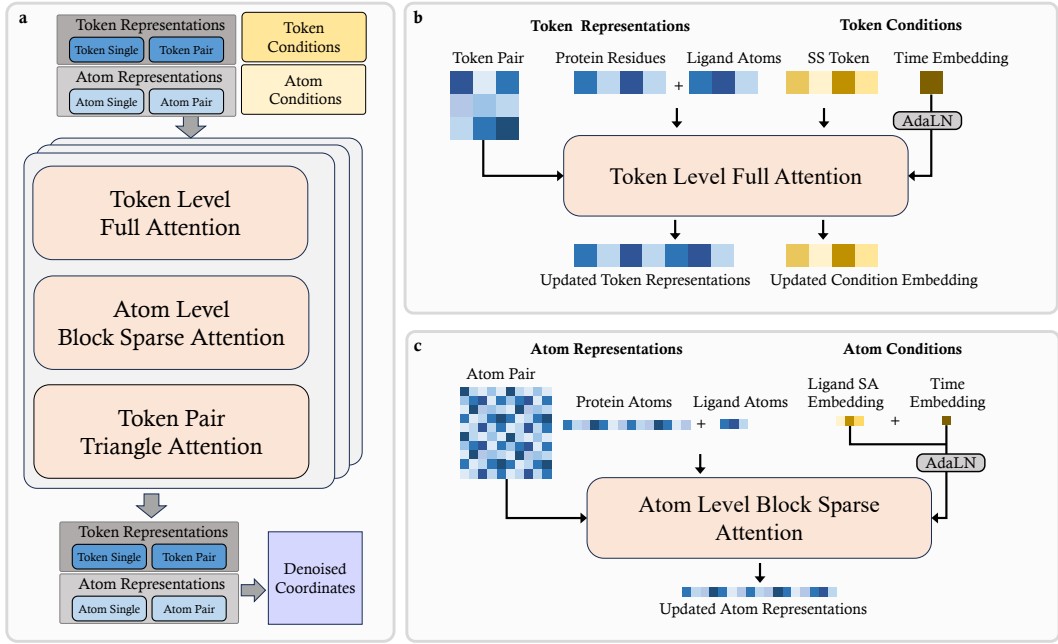

Figure 2: **Pallatom-Ligand Transformer Architecture** (a) The core diffusion transformer consists of three attention modules (light pink). These modules jointly update token- and atom-level representations (dark and light blue), guided by their corresponding conditions (dark and light yellow). The denoised coordinates (purple) are decoded from the final atom representations. (b) The token-level full attention module (light pink) updates the token-level single representations (blue) and condition embedding (yellow). It integrates information from the token-level pair features (left) and the time embedding (right), with black arrows indicating information flow. (c) Similary, the atom-level block sparse attention module (light pink) updates the atom-level single representations (blue). This update is conditioned on the atom pair features (left) as well as the atom-level ligand SA and time embeddings (right).

representation (Qu et al., 2024). Thus, a protein-ligand complex with a $L$-residue protein and a $l$-atom ligand is encoded as a chemical system with $14L + l$ atoms. General atomic features for both small molecules and proteins are initialized by the atom's element number, partial charge, etc. (see the full list in Appendix A.6). Specifically, a per-atom ligand solvent accessibility(SA) feature is fused to ligand atom representations for the purpose of conditional generation through self attention (Section 3.3).

To capture the overall spatial features of the complex structure, we use a coarse-level abstract token representation, following the approach of AlphaFold3 (Abramson et al., 2024). Every 14 atoms in the protein are aggregated into one token, while each ligand atom is kept as an individual token to emphasize the atomic interactions across the protein-ligand interface.

Within this unifying scheme, the atom representation learns the fine-grained heterogeneity unique to each atom in the system, where the ligand and the protein are treated as equal entities. Meanwhile, the hybrid token representation places an emphasis onto ligand atoms and learns the coarse-level structural features sensitive to ligand conformational changes.

## 3.2 PALLATOM-LIGAND TRANSFORMER ARCHITECTURE

We set out to design a network that not only captures the overall structural features of the protein-ligand complex but also learns the interatomic interactions across the protein-ligand interface. More importantly, the network design should enable a mechanism for information exchange between atom-level interactions and token-level protein backbone generation. With these considerations, we designed the triple-attention transformer architecture to jointly process token- and atom-level representations (Figure 2).

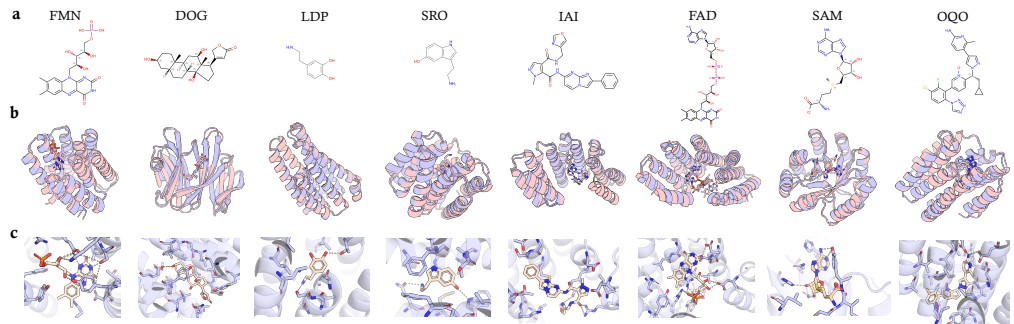

Figure 3: **Ligand-Binding Proteins Designed by Pallatom-Ligand** (a) The chemical structures of eight small molecules are labeled with their three-letter CCD codes. (b) Examples of designed complexes for each target (light purple) are aligned to their AlphaFold3 predictions (light pink). (c) Zoomed-in binding interfaces highlight the hydrogen bonds and electrostatic attractions (yellow dashed lines).

The overall network architecture of Pallatom-Ligand is based on Pallatom (Qu et al., 2024), with several key modifications. Following the Diffusion Transformers (DiT) design philosophy (Peebles & Xie, 2023; Esser et al., 2024; Geffner et al., 2025b), we streamlined the original Pallatom network by replacing its traversing mechanism with a modern transformer architecture. This new transformer operates with three core attention modules: token-level full attention, atom-level block sparse attention, and token pair triangle attention (Figure 2a). To validate the necessity of this triple-stream design, we conducted an ablation study removing individual attention components. The results, detailing the contribution of each stream to the overall generative capability, are provided in Appendix A.16.1.

Denote the token-level single representation as $a \in \mathbb{R}^{(L+l) \times C_{tok}}$, the secondary structure conditions as $\mathrm{cond}_{ss} \in \mathbb{R}^{k \times C_{tok}}$ (where $k = 4$ is the number of secondary structure (SS) tokens), the time embedding as $t \in \mathbb{R}^{C_{t_{emb}}}$, the atom-level single representation as $q \in \mathbb{R}^{(14L+l) \times C_{atom}}$, the atom-level conditions as $c \in \mathbb{R}^{(14L+l) \times C_{atom}}$, the token-level pair representation as $z \in \mathbb{R}^{(L+l) \times (L+l) \times C_{pair}}$ and the atom-level pair representation as $p \in \mathbb{R}^{(14L+l) \times (14L+l) \times C_{atompair}}$. The initialization of these features are described in Appendix A.6.

Each transformer block performs the following operations sequentially:

**Token-level Full Attention** This module takes the token-level representation $a$ and condition $\mathrm{cond}_{ss}$ as input. It updates them with attention bias from token-level pair features $z$, with adaptive layer normalization controlled by time embedding $t$ (Figure 2b):

$$[a, \mathrm{cond}_{ss}] = [a, \mathrm{cond}_{ss}] + \mathrm{Attention}([a, \mathrm{cond}_{ss}], t, z), \tag{1}$$

$$[a, \mathrm{cond}_{ss}] = [a, \mathrm{cond}_{ss}] + \mathrm{Transition}([a, \mathrm{cond}_{ss}], t). \tag{2}$$

This is followed by a token-to-atom conversion:

$$q = q + \mathrm{Layernorm}(a_{tok\_to\_atom}) \tag{3}$$

where $a_{tok\_to\_atom}$ is indexing operation which puts token to their corresponding atoms.

**Atom-level Block Sparse Attention** This module operates directly on the atom representation $q$ with the atom-level conditions $c$, using adaptive layer normalization(controlled by $t$) and atom-level pair features $p$ as attention bias (Figure 2c):

$$q = q + \mathrm{SparseAttention}(q, c, p) \tag{4}$$

$$q = q + \mathrm{Transition}(q, t), \tag{5}$$

This is followed by an atom-to-token conversion:

$$a = a + \mathrm{Layernorm}(\mathrm{SegmentMean}(q)), \tag{6}$$

where SegmentMean is to mean atom features to token level based on their corresponding token index.

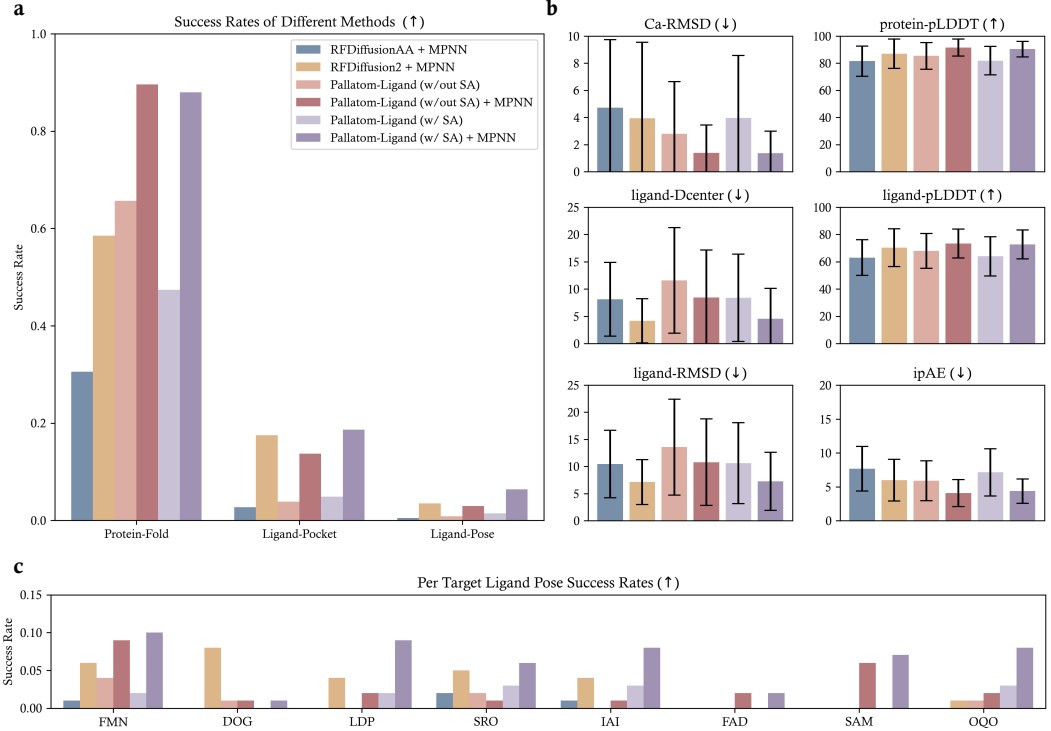

Figure 4: **Benchmark Performance of RFDiffusionAA, RFDiffusion2 and Pallatom-Ligand** (a) Overall success rates are shown for protein fold, ligand pocket, and ligand pose generation. The color scheme is provided in the legend, with full experimental details in Section 4.2 (b) Detailed metrics provide component-wise breakdown comparisons of protein scaffold (top row), ligand pose (middle row), and binding interface (bottom row). (c) Ligand-pose success rates for each of the eight benchmark ligands demonstrate method consistency.

**Coordinate Decoding and Token Pair Triangle Attention** The network decodes the 3D coordinates $r$ for each atom directly from the updated atom representations $q$. Subsequently, the token pair representation $z$ is updated by a token triangle attention module that incoorporates distance information between the center atoms of the tokens:

$$z_{rbf} = \text{Linear}(\text{dist}(r_{center})) \tag{7}$$
$$z = z + \text{TriangleAttention}(z, z_{rbf}, t), \tag{8}$$

The atom pair representation is also updated using the pairwise atom distances via a linear layer.

The detailed implementations of the "Attention" in Eq. 1, "SparseAttention" in Eq. 4, "TriangleAttention" in Eq. 8 and "Transition" in Eq. 2,5 are described in Appendix A.6.

### 3.3 CONDITIONS ON GLOBAL AND ATOMIC SCALES

Leveraging the modular design of our network architecture (Section 3.2), we experimented with novel conditioning strategies at both the token and atom levels. These strategies enable global control over protein fold generation and local control over ligand solvent accessibility, respectively.

**Global Conditioning: Protein Fold Via Alpha Ratio** Natural proteins exhibit a wide variety of tertiary structures with diverse $\alpha/\beta$ compositions, which underpins their vast array of functions in biological systems (Orengo et al., 1997). To design proteins with novel functions, it is therefore essential to explore the protein structure space comprehensively. However, several generative models have been observed to favor all-helical structures (Watson et al., 2023; Geffner et al., 2025b).

| | Protein Scaffold | | | | Ligand Pose | | | Binding Interface |
|---|---|---|---|---|---|---|---|---|
| | Ca-RMSD (↓) | pro-pLDDT (↑) | Div. (↑) | Nov. (↓) | lig-D_center (↓) | lig-RMSD (↓) | lig-pLDDT (↑) | ipAE (↓) |
| RFdiffusionAA (mpnn1) | 4.72 ± 5.02 | 81.52 ± 11.17 | 0.21 | 0.74 | 8.11 ± 6.77 | 10.44 ± 6.23 | 63.06 ± 13.04 | 7.67 ± 3.28 |
| RFdiffusion2 (mpnn1) | 3.94 ± 5.48 | 87.02 ± 10.40 | **0.30** | **0.80** | **4.17 ± 3.97** | 7.12 ± 3.98 | 70.35 ± 12.60 | 7.98 ± 2.08 |
| Ours (w/out SA) | 2.80 ± 3.84 | 84.48 ± 9.84 | 0.17 | 0.85 | 11.56 ± 9.68 | 13.58 ± 8.85 | 67.95 ± 12.73 | 5.90 ± 2.94 |
| Ours (w/out SA) (mpnn1) | 1.39 ± 2.05 | **91.55 ± 6.33** | 0.17 | 0.85 | 8.45 ± 8.70 | 10.78 ± 7.99 | **73.40 ± 10.61** | **4.07 ± 1.98** |
| Ours (w/ SA) | 3.95 ± 4.61 | 81.87 ± 10.45 | 0.14 | 0.81 | 8.37 ± 8.01 | 10.59 ± 7.47 | 64.00 ± 14.43 | 7.13 ± 3.49 |
| Ours (w/ SA) (mpnn1) | **1.36 ± 1.63** | 90.41 ± 5.67 | 0.14 | 0.81 | 4.56 ± 5.56 | **7.04 ± 5.37** | 72.78 ± 10.65 | 4.35 ± 1.80 |

Table 1: **Benchmark Performance on Designing Ligand-Binding Proteins** The best-performing model is indicated in **Bold**, and the second-best is Underlined. Div. and Nov. denote Diversity and Novelty, respectively.

To mitigate such bias, we introduce the "$\alpha$ ratio" as a coarse-level control signal for balancing protein fold generation. This ratio is defined as the number of residues in $\alpha$-helices divided by the total number in $\alpha$-helices and $\beta$-strands, ranging from 0-1. We implemented this global control over protein tertiary structures via three steps: (i) Calculating a per-residue secondary-structure label. (ii) Calculating the $\alpha$ ratio for each training sample and categorizing it into three groups: mainly $\beta$ (0-0.2), $\alpha/\beta$ mix (0.2-0.8), and mainly $\alpha$ (0.8-1). (iii) Injecting this categorical signal into the diffusion process via a concatenated self-attention mechanism at the token level (Figure 2b).

**Atomic Conditioning: Ligand Solvent Accessibility** Controlling ligand solvent accessibility (SA) is crucial in designing ligand-binding proteins for real-world applications. The decision to expose or bury specific ligand atoms within the binding pocket directly impacts the protein's utility as biosensors, therapeutics, and diagnostics.

To enable atom-level control over ligand solvent accessibility, we implemented two key modifications: (i) We quantified the solvent accessibility of each ligand using Relative Solvent Accessibility(RSA) (Ahern et al., 2025), which was then discretized into three categories: fully buried (0–0.1), parittly buried (0.1–1.0), and fully exposed (1.0). (ii) This discrete RSA feature was incorporated as an atom-level input via projected learnable embeddings, allowing it to directly influence structure generation at atomic resolution (Figure 2c).

## 3.4 TRAINING STRATEGIES

**Data Sampling** Compared to image or natural language data, high-quality protein structure data are not only limited by quantity but also suffer from severe distributional imbalance. This imbalance is particularly pronounced in protein-ligand complexes. Some ligands are observed only with specific protein folds, whereas other folds can bind to a wide variety of ligands (Durairaj et al., 2024). Conventional data sampling strategies exacerbate this issue. Clustering samples by overall protein structure leads to uneven ligand frequencies, causing large performance discrepancies across different targets. Conversely, clustering the samples solely by ligand–protein interface results in model collapse with converged protein structures (Appendix A.17).

To address this dilemma caused by data sparsity, we introduced a dual-objective training framework with distinct data sampling strategies: (i) *Learning protein folds*: Samples are selected from structure-based clusters. In this training mode, we apply sequential and spatial cropping to preserve the whole-protein structural context. (ii) *Learning protein-ligand interactions*: Samples are selected via ligand-based clustering. In this mode, we crop the local region around the ligand, allowing the model to focus on atomic interactions independent of the global fold. We use a $1 : 1$ sampling ratio between *Mode* (i) and *Mode* (ii) to balance the objectives, ensuring each ligand type was sampled equally without oversampling specific protein folds. We justify the choice of our balanced sampling strategy through a comparative analysis against fold-centric and interaction-centric baselines. Detailed ablation results demonstrating the necessity of this dual-objective approach for generalizing to rare ligands are presented in Appendix A.16.2.

**Condition Sampling** To ensure compatibility when combining multi-level conditions, we implemented a hierarchical sampling strategy. For the global fold condition($\alpha$ ratio), the label is provided with probability of $p = 0.5$ and omitted otherwise ($p = 0.5$). For the atomic-level condition (ligand SA), we apply a two-tiered strategy: (1) All SA labels are dropped with $p = 0.5$; (2) With $p = 0.25$, labels are provided for all ligand atoms; (3) With $p = 0.25$, labels are provided for a random subset of ligand atoms, where each atom is included independently with $p = 0.5$.

## 4 EXPERIMENTS

### 4.1 METRICS

**AlphaFold3-Based Component-Specific Metrics** Following the established *in silico* protocol of ranking generative models by their consistency with a structure prediction model, existing ligand-binding protein design methods are evaluated using overall AlphaFold3 confidence scores (Cho et al., 2025). We find these aggregate scores insufficient because they fail to reflect the multi-faceted nature of the design challenge. A successful ligand-binding design requires three components: a stable protein scaffold, accurate ligand positioning, and a complementary binding interface; failure in any one aspect results in a non-functional design. To enable a more discriminating evaluation that pinpoints methodological strengths and shortcomings, we introduce a set of component-specific metrics derived from the alignment of the design models to their AlphaFold3 predictions[1]:

- **Protein Scaffold:** RMSD of protein backbone $C_\alpha$ atoms (Ca-RMSD), and the averaged pLDDT over all protein atoms (protein-pLDDT).
- **Ligand Pose:** Distance between ligand centroids (ligand-$D_{center}$), RMSD of ligand atoms (ligand-RMSD), and the average pLDDT over ligand atoms (ligand-pLDDT).
- **Binding Interface:** Interface predicted aligned error (ipAE).

Based on these numerical metrics, we further define three types of success with increasingly stringent criteria:

- **Protein-Fold Success:** Ca-RMSD $< 2\text{Å}$ and protein-pLDDT $> 80$.
- **Ligand-Pocket Success:** Protein-Fold Success and ligand-$D_{center} < 4\text{Å}$ and ligand-pLDDT $> 80$.
- **Ligand-Pose Success:** Protein-Fold Success and ligand-RMSD $< 2\text{Å}$.

In all the tests below, we use these metrics to evaluate the performance of RFdiffusionAA, RFdiffusion2 and Pallatom-Ligand. For a detailed description of all metrics used in this study, please refer to Appendix A.10.

### 4.2 LIGAND-BINDING PROTEIN GENERATION BENCHMARK

We benchmarked Pallatom-Ligand against RFidffusionAA and RFdiffusion2 on a set of eight small molecules selected for their diverse chemical properties (Figure 3, Appendix A.8). To our knowledge, this represents the most comprehensive benchmark for ligand-binding protein design, expanding upon previous four-target set to include molecules with smaller size, opposite charge, and hydrophobic moieties. For each target, we generated 100 protein structures per method at a fixed length. We then designed a single sequence for each structure using LigandMPNN (Dauparas et al., 2025) and evaluated the resulting structure-sequence pairs with our AlphaFold3-based metrics (Experiments 4.1). Full benchmarking protocols are provided in Appendix A.9.

Since Pallatom-Ligand co-generates structures and sequences in its all-atom framework, we used LigandMPNN to redesign only the sequences outside the ligand-binding interface, applying a 6Å atom distance threshold. We evaluated both Pallatom-Ligand's raw sequences and the LigandMPNN-redesigned sequences. For fair comparison with RFdiffusion2 — which was conditioned on a globally buried ligand by default (Appendix A.9) — we performed additional Pallatom-Ligand inferences using our ligand SA control, setting the SA to 0 (fully buried) for all targets. In Figure 4, these four sets of inferences are denoted as: Pallatom-Ligand (w/out SA), Pallatom-Ligand (w/out SA) + MPNN, Pallatom-Ligand (w/ SA), and Pallatom-Ligand (w/ SA) + MPNN. All sampling parameters are listed in Appendix A.7.

In our benchmark, Pallatom-Ligand achieved higher *in silico* success rates than RFdiffusionAA and RFdiffusion2, outperforming them across most of evaluated metrics (Figure 4, Table 1). To elucidate

---

[1]We emphasize that these metrics quantify *structural consistency* only — a necessary condition for valid design and they do not guarantee biological activity.

the source of this improvement, we leverage our component-specific metrics to analyze the results from two complementary perspectives:

**All-Atom Generation of Protein-Ligand Complexes** We first focus on Pallatom-Ligand's raw sequences (light pink and light purple bars in Figure 4) to assess its capability for end-to-end, all-atom generation. In protein fold generation, Pallatom-Ligand's raw sequences achieve high performance, as indicated by the protein-fold success rate (first column of Figure 4a) and individual metrics, including Ca-RMSD and protein-pLDDT (first row of Figure 4b). When explicitly conditioned to bury all ligand atoms, our raw sequences exhibit a trade-off: a decrease in fold success rate alongside a slight increase in ligand-pocket and ligand-pose success rates (light purple vs. light pink bars, Figure 4a). This inverse relationship between protein stability and function is a well-established principle in protein science (Shoichet et al., 1995; Bloom et al., 2006; Tokuriki et al., 2008), and it is encouraging to see our data-driven model recapitulates this trend. Regarding ligand-binding capability, Pallatom-Ligand's raw sequences surpass RFdiffusionAA in both ligand-pocket and ligand-pose success rates (light pink vs. blue bars, second and the third columns, Figure 4a). In addition, our model achieves the highest ligand-pLDDT and the lowest ipAE among all baselines (Figure 4b), underscoring the advantages of joint all-atom modeling of ligands and proteins.

**Advantages of All-Atom Modeling with LigandMPNN** By examining the performance of Pallatom-Ligand's LigandMPNN-redesigned sequences, we isolate the improvements specifically due to all-atom structure generation. Pallatom-Ligand with LigandMPNN achieves the highest *in silico* success rates and the best metrics for both protein folding and ligand binding (dark purple and rosy brown bars, Figure 4). This leading performance stems from a substantial improvement in protein fold quality, complementedy by a consistent improvement in ligand binding performance. Furthermore, Pallatom-Ligand with LigandMPNN demonstrates balanced performance across all eight targets (purple bars, Figure 4c), indicating that our training strategy is well calibrated (Method 3.4). Notably, Pallatom-Ligand successfully generated *in silico* binders for all targets, whereas the RFdiffusion2 failed for FAD and SAM, and RFdiffusionAA succeeded for only three out of eight (Figure 4c).

In summary, this comprehensive benchmark yields two key insights: it not only demonstrates Pallatom-Ligand's superior *in silico* performance over previous state-of-the-art methods but also, by deconvoluting the overall improvement into specific components, provides clear direction for further work— particularly in refining the learning of atom-level ligand-protein interactions.

## 4.3 Conditional Generation with $\alpha$ ratios

Inspired by semantic conditioning in image generation (Rombach et al., 2021; Labs et al., 2025), our model is trained to freely assemble secondary structure elements in order to match a target $\alpha$ ratio, thereby encouraging diversity in protein tertiary structures (Section 3.3).

To validate this strategy, we conducted experiments generating proteins across a spectrum of $\alpha$ ratios and evaluated their ligand-binding success rates as well as secondary structure distributions. For each target, we generated 40 samples under three different secondary structure conditions. Table 2 reports the mean values of these metrics over all eight targets, evaluating Pallatom-Ligand's raw sequences.

|  | Protein-Fold | Ligand-Pocket | Ligand-Pose | Div. | Nov. | $\alpha$% / $\beta$% |
|---|---|---|---|---|---|---|
| RFdiffusionAA (mpnn1) | 30.6% | 2.8% | 0.5% | 0.21 | 0.74 | 62.4/12.1 |
| RFdiffusion2 (mpnn1) | 58.5% | 17.5% | 3.5% | 0.30 | 0.80 | 64.5/14.6 |
| w/out cond. | 71.5% | 4.4% | 1.0% | 0.17 | 0.85 | 79.4 / 3.5 |
| $\alpha \in [0\sim0.2]$ | 56.2% | 2.8% | 0.8% | 0.17 | 0.83 | 9.4 / 57.2 |
| $\alpha \in [0.2\sim0.8]$ | 60.8% | 6.3% | 0.9% | 0.26 | 0.85 | 61.7 / 19.4 |
| $\alpha \in [0.8\sim1.0]$ | 66.8% | 11.0% | 1.5% | 0.15 | 0.89 | 82.4 / 0.5 |

Table 2: **Global Control of Protein Fold** We report three success rates, diversity, novelty and the average $\alpha/\beta$ ratios in our generated samples.

Our model correctly follows the conditioning signals, successfully producing proteins with distinct secondary structure preferences, as shown in the $\alpha$% / $\beta$% columns of the table. These preferences further influence the design success rates, where a higher proportion of $\beta$ structures corresponds to lower success rates. Moreover, the setting of $\alpha$ ratios (0.2–0.8) achieves the highest diversity and novelty, as it covers the broadest range. Figure 6 illustrates the diverse structures of Pallatom-Ligand's output under different SS control.

## 4.4 CONDITIONAL GENERATION WITH LIGAND SOLVENT ACCESSIBILITY

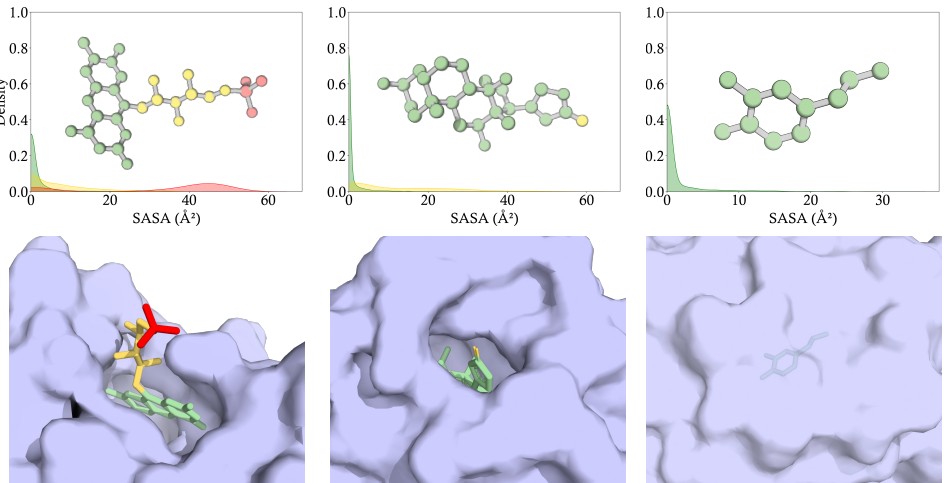

Figure 5: **Atomic Control of Ligand Solvent Accessibility** The **top** panel shows three selected small molecules with predefined ligand SA labels. Fully buried atoms are colored mint (SA label = 0), half buried atoms are colored yellow (SA label =0.5), and fully exposed atoms are colored in red (SA label =1). The distribution of ligand solvent accessible surface area (SASA) across forty designs is plotted for each small molecule. The **bottom** panel shows zoomed-in views of representative binding pockets, with the protein surface contour displayed in purple.

To further evaluate the model's capacity for conditional generation at the atomic level, we designed proteins to bind specific ligands with predefined solvent accessibility (SA) labels. We defined per-atom SA labels for three target ligands (FMN, DOG and LDP in Figure 5) based on their chemical properties and binding patterns in natural proteins. For each ligand, we generated 40 proteins of a fixed length. The distribution of the ligand's solvent-accessible surface area (SASA) in the generated proteins (Figure 5) aligns closely with the design control, confirming that the model produces proteins with suitable binding pockets for all three tasks.

## 5 CONCLUSIONS

We present Pallatom-Ligand, an all-atom diffusion model for the end-to-end generation of protein-ligand complexes. Conditioned solely on a small-molecule target, our model generates complete complex structures with atomic detail for the protein backbone and side chains. In a comprehensive *in silico* benchmark, Pallatom-Ligand outperforms the previous state-of-the-art, RFDiffusionAA and RFDiffusion2, across a wide range of metrics. Furthermore, the all-atom framework enables more expressive design conditions at both atomic and residue levels, as demonstrated by the model's control over the protein fold generation and ligand solvent accessibility.

By enabling end-to-end design of ligand-binding proteins with minimal prior assumptions, Pallatom-Ligand opens new avenues for exploring protein function space. Its strong *in silico* performance and expressive controllability make it directly applicable to biosensor development and other biotechnological applications.

While Pallatom-Ligand represents a significant advance, future work will focus on two key directions. First, we aim to broaden the scope of supported biomolecular complexes beyond protein–small molecule systems, including nucleic acid–containing assemblies, covalent ligand binding, and proteins with noncanonical amino acids. Second, we plan to scale training to larger distilled protein-ligand datasets (Lemos et al., 2025), in order to further push the performance limits of all-atom ligand binding design.

## 6 Acknowledgement

We thank Yifan Qin, Ge Tian, Ruiyi Zhang, Jiakai Zhang and anonymous ICLR reviewers for helpful feedback and discussions, and the ShanghaiTech HPC platform for generous technical support. This work was supported by 2024 Shanghai Action for Science, Technology and Innovation Program of Natural Science Foundation of Shanghai (24JS2820100).

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

# A APPENDIX

## CONTENTS

## A.1 ETHICS STATEMENT

This work focuses on developing machine learning methods for protein–ligand complex generation. Our study does not involve human subjects, animal experiments, or private or sensitive data. All datasets used in this work are publicly available from prior publications and are widely used within the structural biology and machine learning communities. The research does not introduce new safety, security, or ethical risks beyond those already associated with existing protein design methodologies. We acknowledge that downstream applications of protein design may carry potential dual-use concerns; however, our contributions are methodological and intended for advancing fundamental research.

## A.2 REPRODUCIBILITY STATEMENT

We have taken several steps to ensure the reproducibility of our work. All datasets used in our experiments are publicly available, and the details of dataset preprocessing are provided in Appendix A.4. The full model architecture, training setup, and hyperparameter choices are described in Section 3.2 and Appendices A.6, A.14, and A.15. Evaluation protocols are included in Appendix A.9. Sampling configurations are included in Appendix A.7.

## A.3 THE USE OF LARGE LANGUAGE MODELS

Large language models (LLMs) were used solely for editorial purposes in this work, specifically to polish language and improve the clarity of writing. They were not involved in generating ideas, designing methods, conducting experiments, or analyzing results. The scientific contributions and findings presented in this paper are entirely the work of the authors.

## A.4 DATA CURATION

**Protein-Ligand Complexes from PDB.** We curated our protein-ligand dataset based on the methodology of AlphaFold3 (Abramson et al., 2024). Each biological assembly was treated as a single data point; for PDB entries containing multiple assemblies, only the first was used. The following filtering criteria were applied:

- Experimental Method: Structures determined by X-ray diffraction or electron microscopy.
- Resolution: A resolution better than 3 Å.
- Sequence Length: A minimum of 32 residues.

- Complex Size: Complexes larger than 7 MB were excluded.
- Clashing Chains: Chains with more than 30

This procedure resulted in a final dataset of 66,201 protein-ligand complexes. **Protein Structure Distillation from AFDB.** Following recent works (e.g., Genie2 (Lin et al., 2024), Proteina (Geffner et al., 2025b), and Pallatom (Qu et al., 2024)), we trained our model on synthetic protein structures from the AlphaFold Database (AFDB). To ensure the quality of these distilled structures, we applied the following filters:

- Quality and Length: An average pLDDT score greater than 80 and a maximum sequence length of 256 residues.
- Minimum Length: A minimum sequence length of 32 residues.
- Disordered Regions: Disordered regions at the N- and C-termini were trimmed.
- Redundancy Removal: Redundant structures were removed using the FoldSeek easy-cluster algorithm, with a TM-score threshold of 0.7 and a sequence coverage of 0.9.

Ultimately, this process yielded a set of 495,322 protein monomers.

## A.5  VISUALIZATION OF FOLD CONDITIONED SAMPLES

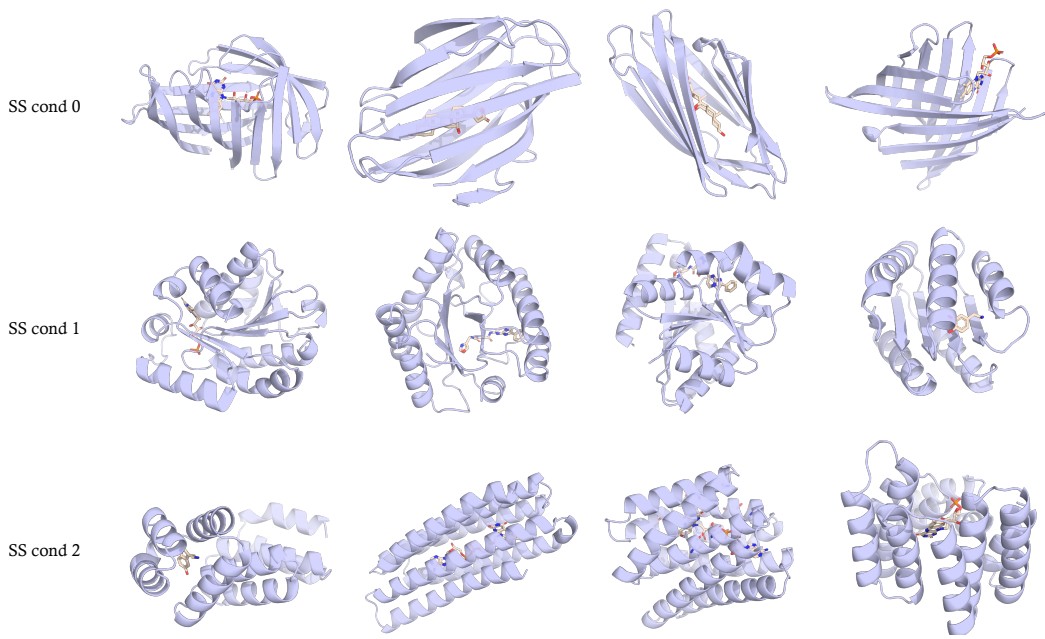

Figure 6: **Fold Conditioned Ligand-Binding Proteins Designed by Pallatom-Ligand via different $\alpha$ ratio** SS cond 0 refers to $\alpha$ ratio ranging from 0-0.2, SS cond 1 refers to $\alpha$ ratio ranging from 0.2-0.8 and SS cond 2 refers to $\alpha$ ratio ranging from 0.8-1.0.

## A.6  ARCHITECTURE DETAILS

We use the following channel dimensions: 384 for tokens, 256 for atoms, 128 for token pairs, 16 for atom pairs, and 32 for the time embedding. For the initialization of our representations and condition embeddings, we encode the following information:

- Reference conformer positions, element types, charges, degree information, atom indices, and SASA features into the **atom condition embedding** $c$.
- Noisy coordinates, the atom condition embedding, and the token condition embedding into the **atom representation** $q$.

- The aggregated atom representation into the **token representation** $a$.

- Relative token/residue/chain indices and the token pairwise distogram of noisy coordinates into the **token pair representation** $z$.

- Atom pairwise distances in the reference conformer, relative atom positions, and the expanded token pair representation into the **atom pair representation** $p$.

For token-level attention, we use 8 attention heads, a dropout rate of 0.15, and no query-key (QK) bias. For atom-level attention, we also use 8 attention heads and a dropout rate of 0.15, but with QK bias enabled. In the triangle block, we use 4 attention heads, a dropout rate of 0.15, and no QK bias.

The 'Attention' mechanism in Eq. 1 and the 'SparseAttention' mechanism in Eq. 4 share the same fundamental calculation. The only difference is the application of a $32 \times 128$ block mask in 'SparseAttention'. The detailed implementation is provided in Algorithm 1.

---

**Algorithm 1** Attention with Pair Bias

---

**Input:** Single representation $x$, padding mask $x_{\text{mask}}$, pair representation $y$, condition embedding $cond$, (optional) attention mask $\beta$.

1:

**Output:** Update to the single representation $\Delta x$.

    # Layer normalization of inputs

2: $x \leftarrow \text{AdaLN}(x, cond)$         ▷ Adaptive Layer Normalization

3: $y \leftarrow \text{LayerNorm}(y)$

    # Compute pair bias from pair representation

4: $W_{\text{bias}} \in \mathbb{R}^{C_y \times H} \leftarrow$ Initialized weights     ▷ $H$ is the number of heads

5: $\text{bias} \leftarrow \text{Einsum}('qkc, ch \rightarrow hqk', y, W_{\text{bias}})$     ▷ Pair bias for attention scores

    # Prepare attention mask

6: $\text{padding\_mask} \leftarrow (1 - x_{\text{mask}}) \times -10^9$     ▷ Convert binary mask to attention bias

7: $\text{attention\_mask} \leftarrow \beta * \text{padding\_mask}$     ▷ Convert binary mask to attention bias

    # Multi-Head Attention with pair bias

8: $\Delta a \leftarrow \text{MultiHeadAttention}(\text{query} = x, \text{key} = x, \text{value} = x, \text{bias} = \text{bias}, \text{mask} = \text{attention\_mask})$

    # Apply output gate

9: $\text{gate} \leftarrow \sigma(\text{Linear}_{\text{gate}}(cond))$     ▷ $\sigma$ is the sigmoid function

10: $\Delta x \leftarrow \Delta x \odot \text{gate}$     ▷ Element-wise product

11: **return** $\Delta x$

---

For "Attention" (Eq. 1), the inputs to the algorithm are $x = [a, \text{cond}_{ss}]$, $y = z$, and $cond = t$. For "SparseAttention" (Eq. 4), the inputs are $x = q$, $y = p$, and $cond = c$, with an additional block mask input $\beta$ of size $32 \times 128$. This setup is consistent with that of AlphaFold3 (Abramson et al., 2024) and Pallatom (Qu et al., 2024).

The details of the "TriangleAttention" mechanism (Eq. 8) are described in Algorithm 2.

The "Transition module," referenced in Eq. 2 and Eq. 5, is a conditioning-aware feed-forward network designed to update an input representation **a** based on a conditioning vector **s**. The details are described in Algorithm 3.

## A.7 SAMPLING PARAMETERS

Our sampling scheme is adopted from Pallatom (Qu et al., 2024), and we keep the notation consistent for clarity. The hyperparameters were set as follows: an initial noise level of $\gamma_0 = 0.4$, a step scale of $\eta = 1.75$, and a total of $N = 500$ sampling steps. We utilized the same noise schedule as the original work without further tuning. The inference time is reported in Appendix A.12.

## A.8 SMALL-MOLECULE TARGET DESCRIPTION

**IAI** ($C_{22}H_{18}N_8O_3$) is a mid-sized (442.4 Da), neutral polar molecule with balanced hydrophobicity. Its hydrogen-bonding capacity includes 2 donors and 7 acceptors. As a phosphodiesterase inhibitor,

---

**Algorithm 2** Conditional Triangle Attention

---

**Input:** Pair activations $z$, rbf activations $z_{\text{rbf}}$, condition embedding $t$, pair mask $m$.
**Output:** Updated pair activations $z'$.
    # Normalize inputs.
1: $z \leftarrow \text{LayerNorm}(z)$
2: $z_{\text{rbf}} \leftarrow \text{AdaLN}(z_{\text{rbf}}, t, \text{use\_bias=True})$
    # Construct attention bias from pair mask.
3: $\text{bias} \leftarrow (m - 1) \times 10^9$
4: $\text{bias} \leftarrow \text{ExpandDims}(\text{bias}, \text{axis} = [1, 2])$               $\triangleright$ shape $[N, 1, 1, N]$
    # Compute non-batched bias from affine activations.
5: Initialize weights $W_{\text{2d}} \in \mathbb{R}^{C_z \times H}$ with std $1/\sqrt{C_z}$
6: $\text{nonbatched\_bias} \leftarrow \text{Einsum}('qkc, ch \rightarrow hqk', z_{\text{rbf}}, W_{\text{2d}})$
    # Multi-head attention with affine bias.
7: $z' \leftarrow \text{MultiHeadAttention}(q = z, k = z, v = z,$
                           $\text{bias} = \text{bias}, \text{extra\_bias} = \text{nonbatched\_bias})$
8: **return** $z'$

---

**Algorithm 3** The Transition Module

---

**Input:** Input representation $\mathbf{a}$, conditioning vector $\mathbf{s}$.
**Output:** Updated representation $\mathbf{a}_{\text{out}}$.
    # 1. Adaptive Layer Normalization.
1: $\gamma, \beta \leftarrow \text{Linear}(\mathbf{s}), \text{Linear}(\mathbf{s})$          $\triangleright$ Produce gain and bias from condition $\mathbf{s}$.
2: $\mathbf{a}_{\text{norm}} \leftarrow \gamma \odot \text{LayerNorm}(\mathbf{a}) + \beta$          $\triangleright \odot$ denotes element-wise product.
    # 2. Gated Feed-Forward Layer (SwiGLU).
3: $\text{gate} \leftarrow \text{Swish}(\text{Linear}_1(\mathbf{a}_{\text{norm}}))$
4: $\text{val} \leftarrow \text{Linear}_2(\mathbf{a}_{\text{norm}})$
5: $\mathbf{b} \leftarrow \text{gate} \odot \text{val}$
    # 3. Final Gating based on condition.
6: $\text{output\_gate} \leftarrow \text{Sigmoid}(\text{Linear}_3(\mathbf{s}))$
7: $\text{output\_val} \leftarrow \text{Linear}_4(\mathbf{b})$
8: $\mathbf{a}_{\text{out}} \leftarrow \text{output\_gate} \odot \text{output\_val}$
9: **return** $\mathbf{a}_{\text{out}}$

---

IAI has been investigated for treating neurological conditions like schizophrenia and Huntington's disease.

**SAM** (S-Adenosylmethionine) is a moderately large (398.4 Da), neutral molecule with high polarity and strong hydrophilicity, containing 4 hydrogen-bonding donors and 10 acceptors. Its primary biological role is as a universal methyl donor in one-carbon metabolism. This function extends to gene regulation via riboswitches in bacteria and the regulation of nucleic acid methylation and amino acid metabolism in eukaryotic cells.

**OQO** ($C_{26}H_{23}ClFN_9O$) is a relatively large (532 Da), nominally neutral molecule that exhibits charge resonance, enabling ionization in different environments. It displays moderate polarity and balanced hydrophobicity, with a hydrogen-bonding capacity of 1 donor and 8 acceptors. OQO has been investigated as a FXIa inhibitor for treating stroke and myocardial infarction.

**FAD & FMN** Flavin mononucleotide (FMN) and flavin adenine dinucleotide (FAD) are flavin-derived cofactors that serve as prosthetic groups in numerous redox enzymes. Both are derivatives of vitamin B2 (riboflavin), formed via phosphorylation to FMN or further adenylylation to FAD. FMN has a molecular weight of 456.3 Da and contains 6 hydrogen-bond donors and 10 acceptors, indicating strong hydrophilicity. FAD is larger, with a molecular weight of 785.5 Da, 9 hydrogen-bond donors, 20 acceptors, reflecting very high polarity and aqueous solubility. Overall, both molecules are highly polar, mid- to large-sized cofactors with substantial hydrogen-bonding potential, consistent with their roles as enzyme-bound prosthetic groups in redox biochemistry.

**DOG** (Digoxigenin $C_{23}H_{34}O_5$) is a widely used small molecule in biochemical assays. It is typically conjugated to biomolecules, such as proteins or nucleic acids, to enable detection via high-

affinity, specific anti-digoxigenin antibodies. From a physicochemical perspective, DOG is a relatively small (390.5 Da), moderately polar molecule with balanced hydrogen-bonding capacity(3 hydrogen-bond donors and 5 acceptors).

**LDP** (L-Dopamine) is a neurotransmitter that transmits chemical signals between neurons and plays a central role in reward processing. From a physicochemical perspective, LDP is a small (153.18 Da), polar molecule with balanced hydrogen-bonding potential(3 donors and 3 acceptors).

**SRO** (Serotonin 5-HT) is a monoamine neurotransmitter with broad functions in both the central nervous system (CNS) and peripheral tissues. From a physicochemical perspective, 5-HT has a molecular weight of 176.21 Da, contains 3 hydrogen-bond donors and 2 acceptors.

## A.9 Details For Baselines' Evaluation

### A.9.1 RFdiffusion-AA

For the evaluation of RFdiffusion-AA, we closely followed the protocol from the Hem binder design study (Krishna et al., 2024). The procedure consisted of the following steps:

**Conformer Sampling:** We first sampled 10 ligand conformers. This step was necessary because RFdiffusion-AA does not alter the input ligand's conformation.

**Backbone Generation:** For each of the 10 conformers, we designed 10 protein backbones using the default configuration of RFdiffusion-AA, yielding a total of 100 backbones.

**Sequence Design:** Subsequently, for each backbone, we designed a single sequence using LigandMPNN, providing the corresponding ligand structure as conditional input.

**Structure Prediction:** Finally, the structures for all 100 designed sequences were predicted using AlphaFold3 without MSAs. For each sequence, we generated 5 structures using a fixed random seed of 1.

### A.9.2 RFdiffusion2

Our evaluation protocol for RFdiffusion2 followed the `small_molecule_binder_rasa_buried` example from the official `open_source_demo.json` file.

**Backbone Generation:** The initial data preparation involved selecting high-quality protein-ligand complexes for eight distinct small molecules from the Protein Data Bank (PDB). In cases where a protein bound multiple instances of the same ligand, one complex was arbitrarily selected. The geometric center of the ligand's atoms was defined as its origin point (ORI) and included in the PDB file to guide scaffold generation. To promote ligand encapsulation, we adopted the recommended settings from the example, setting the RASA value to 0. Following this procedure, we generated 100 scaffolds for each of the eight ligands.

**Sequence Design & Structure Prediction:** The subsequent steps of sequence design and structure prediction were identical to the protocol used for RFdiffusion-AA: sequences were designed using LigandMPNN, and their corresponding structures were predicted with AlphaFold3.

## A.10 Definition of Evaluation Metrics

To enhance the accessibility of our paper for readers across different backgrounds, we provide detailed definitions of the technical metrics used in our evaluation (Section 4.2).

- **Protein Scaffold Metrics:**
  - **C$\alpha$-RMSD:** The Root-Mean-Square Deviation (RMSD) of the protein backbone C$\alpha$ atoms between the generated structure and the reference ground truth, measured in Ångstroms (Å). Lower values indicate higher structural fidelity to the native conformation.
  - **Protein pLDDT:** The average predicted Local Distance Difference Test (pLDDT) score calculated over all protein residues. This score ranges from 0 to 100 and serves as a confidence metric for the local structural quality predicted by the folding model (e.g., AlphaFold2/ESMFold). Higher values signify greater confidence.

– **Diversity:** We assess the structural diversity of the generated samples by clustering them using Foldseek (van Kempen et al., 2024). The metric reports the total number of distinct clusters formed; a higher count indicates that the model can generate a more diverse set of structural solutions.

– **Novelty:** This metric quantifies the dissimilarity of generated samples relative to known structures in the PDB. For each generated sample, we compute the maximum TM-score (Zhang & Skolnick, 2005) against the entire PDB database. The reported novelty is the average of these maximum TM-scores. Since a higher TM-score implies similarity, a *lower* average score indicates that the generated proteins represent novel folds unseen in the training distribution.

• **Ligand Pose Metrics:**

– **Ligand $D_{center}$:** The Euclidean distance between the geometric centroid of the generated ligand and that of the reference ligand. This metric measures positional consistency, effectively identifying large translational deviations of the generated ligand relative to the binding pocket.

– **Ligand RMSD:** The Root-Mean-Square Deviation of the ligand's heavy atoms between the generated sample and the reference structure. Crucially, this is calculated *after* superimposing the generated protein backbone onto the reference backbone. A smaller value indicates high accuracy in both the internal conformation and the relative orientation of the ligand within the pocket.

– **Ligand pLDDT:** The average pLDDT score calculated over all ligand atoms. Similar to the protein score, this reflects the model's confidence in the predicted coordinates of the ligand.

• **Binding Interface Metrics:**

– **ipAE (Interface Predicted Aligned Error):** This metric estimates the positional error (in Å) between pairs of residues and ligand atoms across the protein-ligand interface. Lower ipAE values indicate higher confidence in the specific interaction geometry and the relative packing of the ligand against the protein surface.

## A.11    DETAILED RESULTS FOR OUR LIGAND-BINDING PROTEIN DESIGN BENCHMARK

### A.11.1    RFDIFFUSIONAA

| | Ca-RMSD | protein-pLDDT | Diversity | Novelty | ligand-Dcenter | ligand-RMSD | ligand-pLDDT | ipAE | Success Rate | | |
| --- | --- | --- | --- | --- | --- | --- | --- | --- | --- | --- | --- |
| | | | | | | | | | Fold | Pocket | Pose |
| FMN | 4.45 ± 4.45 | 80.79 ± 10.77 | 0.18 | 0.74 | 8.21 ± 6.73 | 10.05 ± 6.29 | 61.50 ± 11.24 | 7.71 ± 3.08 | 29.0% | 2.0% | 1.0% |
| DOG | 6.29 ± 6.75 | 79.80 ± 11.33 | 0.57 | 0.79 | 10.58 ± 9.32 | 12.57 ± 8.40 | 62.82 ± 12.04 | 8.29 ± 3.53 | 30.0% | 3.0% | 0.0% |
| LDP | 4.21 ± 4.95 | 84.53 ± 10.72 | 0.33 | 0.73 | 8.56 ± 6.65 | 9.54 ± 6.18 | 70.11 ± 13.42 | 6.40 ± 3.35 | 36.0% | 7.0% | 0.0% |
| SRO | 4.46 ± 4.95 | 82.58 ± 11.93 | 0.28 | 0.75 | 7.88 ± 7.02 | 9.05 ± 6.52 | 67.73 ± 14.69 | 6.98 ± 3.62 | 36.0% | 7.0% | 2.0% |
| IAI | 4.96 ± 4.72 | 79.90 ± 12.20 | 0.29 | 0.73 | 6.44 ± 5.35 | 9.10 ± 5.07 | 60.41 ± 13.71 | 5.79 ± 2.50 | 31.0% | 1.0% | 1.0% |
| FAD | 4.11 ± 3.77 | 80.25 ± 8.81 | 0.13 | 0.73 | 6.42 ± 4.50 | 11.31 ± 4.44 | 56.19 ± 9.40 | 8.34 ± 3.50 | 26.0% | 0.0% | 0.0% |
| SAM | 4.48 ± 4.95 | 80.95 ± 12.00 | 0.15 | 0.73 | 6.59 ± 4.95 | 9.21 ± 4.45 | 63.58 ± 11.41 | 8.30 ± 2.23 | 30.0% | 0.0% | 0.0% |
| OQO | 4.84 ± 4.94 | 83.36 ± 10.60 | 0.32 | 0.75 | 10.23 ± 7.21 | 12.69 ± 6.36 | 62.17 ± 12.95 | 7.57 ± 3.40 | 27.0% | 2.0% | 0.0% |
| Avg. | 4.72 ± 5.02 | 81.52 ± 11.17 | 0.21 | 0.74 | 8.11 ± 6.77 | 10.44 ± 6.23 | 63.06 ± 13.04 | 7.67 ± 3.28 | 30.6% | 2.8% | 0.5% |

### A.11.2    RFDIFFUSION2

| | Ca-RMSD | protein-pLDDT | Diversity | Novelty | ligand-Dcenter | ligand-RMSD | ligand-pLDDT | ipAE | Success Rate | | |
| --- | --- | --- | --- | --- | --- | --- | --- | --- | --- | --- | --- |
| | | | | | | | | | Fold | Pocket | Pose |
| FMN | 4.18 ± 5.57 | 84.43± 11.76 | 0.34 | 0.79 | 4.14 ± 3.32 | 6.47 ± 3.35 | 66.90 ± 14.50 | 6.52 ± 3.33 | 57.0% | 16.0% | 6.0% |
| DOG | 4.38 ± 7.55 | 87.83 ± 11.35 | 0.49 | 0.81 | 3.89 ± 5.51 | 6.76 ± 5.25 | 72.67 ± 13.56 | 5.76 ± 3.59 | 65.0% | 24.0% | 8.0% |
| LDP | 3.46 ± 4.28 | 89.26 ± 10.43 | 0.49 | 0.80 | 4.50 ± 4.63 | 6.09 ± 4.16 | 77.88 ± 12.97 | 4.89 ± 2.80 | 61.0% | 30.0% | 4.0% |
| SRO | 3.39 ± 4.98 | 89.23 ± 9.83 | 0.45 | 0.81 | 4.53 ± 4.64 | 6.11 ± 4.19 | 76.88 ± 12.15 | 5.02 ± 2.75 | 68.0% | 28.0% | 5.0% |
| IAI | 2.68 ± 3.99 | 90.19 ± 7.72 | 0.59 | 0.80 | 3.54 ± 2.93 | 6.95 ± 3.63 | 70.44 ± 10.81 | 5.48 ± 1.97 | 68.0% | 11.0% | 4.0% |
| FAD | 5.00 ± 5.96 | 81.67 ± 10.18 | 0.29 | 0.79 | 3.95 ± 3.15 | 9.47 ± 3.78 | 58.85 ± 11.16 | 7.52 ± 2.71 | 33.0% | 1.0% | 0.0% |
| SAM | 3.75 ± 5.21 | 85.92 ± 11.08 | 0.3 | 0.80 | 4.43 ± 3.73 | 7.63 ± 3.51 | 70.92 ± 13.22 | 6.06 ± 3.32 | 58.0% | 21.0% | 0.0% |
| OQO | 4.67 ± 6.32 | 87.66 ± 10.86 | 0.53 | 0.80 | 4.37 ± 3.82 | 7.47 ± 3.96 | 68.30 ± 12.40 | 6.59 ± 3.06 | 57.0% | 9.0% | 1.0% |
| Avg. | 3.94 ± 5.61 | 87.03 ± 10.76 | 0.30 | 0.80 | 4.17 ± 4.05 | 7.12 ± 4.13 | 70.37 ± 13.79 | 7.98 ± 2.08 | 58.5% | 17.5% | 3.5% |

### A.11.3 PALLATOM-LIGAND (W/OUT SA)

| | Ca-RMSD | protein-pLDDT | Diversity | Novelty | ligand-Dcenter | ligand-RMSD | ligand-pLDDT | ipAE | Success Rate | | |
| | | | | | | | | | Fold | Pocket | Pose |
|---|---|---|---|---|---|---|---|---|---|---|---|
| FMN | 3.54 ± 4.88 | 84.23 ± 10.01 | 0.16 | 0.84 | 11.32 ± 11.20 | 12.97 ± 10.74 | 66.67 ± 12.39 | 6.55 ± 2.86 | 59.0% | 7.0% | 4.0% |
| DOG | 1.95 ± 2.92 | 85.68 ± 8.53 | 0.17 | 0.86 | 12.45 ± 10.35 | 14.16 ± 9.48 | 67.45 ± 10.73 | 5.95 ± 2.77 | 78.0% | 7.0% | 1.0% |
| LDP | 2.07 ± 2.17 | 90.19 ± 5.67 | 0.15 | 0.89 | 12.95 ± 8.48 | 13.65 ± 8.02 | 74.24 ± 9.26 | 4.18 ± 1.78 | 71.0% | 2.0% | 0.0% |
| SRO | 2.25 ± 2.77 | 89.90 ± 6.22 | 0.15 | 0.91 | 12.95 ± 9.41 | 13.88 ± 8.74 | 75.55 ± 8.09 | 4.33 ± 2.00 | 71.0% | 5.0% | 2.0% |
| IAI | 2.39 ± 2.55 | 86.36 ± 8.84 | 0.18 | 0.83 | 12.42 ± 9.23 | 14.93 ± 8.23 | 65.66 ± 11.69 | 5.79 ± 2.50 | 65.0% | 0.0% | 0.0% |
| FAD | 3.94 ± 5.02 | 78.66 ± 12.78 | 0.19 | 0.79 | 9.04 ± 9.05 | 13.06 ± 7.91 | 57.65 ± 13.37 | 8.24 ± 3.22 | 54.0% | 0.0% | 0.0% |
| SAM | 3.07 ± 4.02 | 85.34 ± 10.87 | 0.15 | 0.82 | 10.42 ± 9.43 | 12.65 ± 8.62 | 71.57 ± 12.37 | 5.87 ± 3.16 | 60.0% | 6.0% | 2.0% |
| OQO | 2.76 ± 3.65 | 86.26 ± 9.89 | 0.17 | 0.82 | 10.97 ± 9.72 | 13.34 ± 8.79 | 64.84 ± 13.62 | 6.30 ± 2.92 | 63.0% | 8.0% | 1.0% |
| Avg. | 2.80 ± 3.84 | 84.48 ± 9.84 | 0.17 | 0.85 | 11.56 ± 9.68 | 13.58 ± 8.85 | 67.95 ± 12.73 | 5.90 ± 2.94 | 55.7% | 3.9% | 0.9% |

### A.11.4 PALLATOM-LIGAND (W/OUT SA) + MPNN

| | Ca-RMSD | protein-pLDDT | Diversity | Novelty | ligand-Dcenter | ligand-RMSD | ligand-pLDDT | ipAE | Success Rate | | |
| | | | | | | | | | Fold | Pocket | Pose |
|---|---|---|---|---|---|---|---|---|---|---|---|
| FMN | 1.48 ± 2.72 | 89.89 ± 7.76 | 0.16 | 0.84 | 8.55 ± 9.85 | 10.56 ± 9.38 | 72.60 ± 11.69 | 4.43 ± 2.24 | 93.0% | 15.0% | 9.0% |
| DOG | 1.16 ± 1.43 | 91.45 ± 4.80 | 0.17 | 0.86 | 8.50 ± 9.23 | 10.76 ± 8.32 | 72.79 ± 7.41 | 3.99 ± 1.54 | 91.0% | 14.0% | 1.0% |
| LDP | 1.30 ± 1.93 | 94.62 ± 2.66 | 0.15 | 0.89 | 10.07 ± 8.30 | 11.18 ± 7.59 | 78.87 ± 6.33 | 2.82 ± 1.05 | 94.0% | 17.0% | 2.0% |
| SRO | 1.31 ± 1.06 | 94.45 ± 1.81 | 0.15 | 0.91 | 12.10 ± 9.54 | 13.06 ± 8.88 | 79.77 ± 5.91 | 2.82 ± 0.52 | 85.0% | 11.0% | 1.0% |
| IAI | 1.20 ± 1.22 | 92.60 ± 3.88 | 0.18 | 0.83 | 8.31 ± 8.69 | 11.15 ± 8.05 | 71.82 ± 9.24 | 3.99 ± 1.33 | 90.0% | 14.0% | 1.0% |
| FAD | 1.99 ± 3.34 | 86.06 ± 9.53 | 0.19 | 0.79 | 6.41 ± 7.22 | 10.41 ± 6.89 | 63.52 ± 12.51 | 6.12 ± 2.76 | 91.0% | 4.0% | 2.0% |
| SAM | 1.32 ± 1.88 | 91.46 ± 5.67 | 0.15 | 0.82 | 7.83 ± 8.60 | 10.20 ± 8.02 | 78.04 ± 8.51 | 3.86 ± 1.68 | 93.0% | 19.0% | 6.0% |
| OQO | 1.41 ± 1.71 | 91.72 ± 6.31 | 0.17 | 0.82 | 5.88 ± 6.36 | 8.97 ± 6.01 | 69.83 ± 10.97 | 4.59 ± 1.73 | 90.0% | 16.0% | 2.0% |
| Avg. | 1.39 ± 2.05 | 91.55 ± 6.33 | 0.17 | 0.85 | 8.45 ± 8.70 | 10.78 ± 7.99 | 73.40 ± 10.61 | 4.07 ± 1.98 | 89.6% | 13.8% | 3.0% |

### A.11.5 PALLATOM-LIGAND (W/ SA)

| | Ca-RMSD | protein-pLDDT | Diversity | Novelty | ligand-Dcenter | ligand-RMSD | ligand-pLDDT | ipAE | Success Rate | | |
| | | | | | | | | | Fold | Pocket | Pose |
|---|---|---|---|---|---|---|---|---|---|---|---|
| FMN | 4.51 ± 5.15 | 81.40 ± 10.38 | 0.13 | 0.81 | 7.83 ± 8.05 | 9.62 ± 7.68 | 63.01 ± 12.99 | 7.32 ± 3.23 | 43.0% | 5.0% | 2.0% |
| DOG | 3.60 ± 4.28 | 82.12 ± 7.72 | 0.10 | 0.82 | 8.57 ± 9.06 | 10.83 ± 8.08 | 63.66 ± 10.20 | 6.97 ± 2.91 | 50.0% | 3.0% | 0.0% |
| LDP | 3.09 ± 3.37 | 88.75 ± 7.07 | 0.11 | 0.85 | 8.79 ± 9.68 | 9.93 ± 9.14 | 73.26 ± 10.44 | 4.68 ± 2.42 | 55.0% | 9.0% | 2.0% |
| SRO | 2.98 ± 3.13 | 86.75 ± 7.56 | 0.09 | 0.84 | 7.87 ± 7.51 | 9.02 ± 6.98 | 73.02 ± 10.26 | 5.28 ± 2.53 | 58.0% | 8.0% | 3.0% |
| IAI | 4.67 ± 4.96 | 79.15 ± 12.28 | 0.15 | 0.80 | 8.78 ± 7.52 | 11.37 ± 7.08 | 60.54 ± 16.43 | 7.88 ± 3.67 | 38.0% | 7.0% | 3.0% |
| FAD | 6.94 ± 6.75 | 71.09 ± 11.89 | 0.24 | 0.78 | 9.77 ± 8.13 | 13.03 ± 7.49 | 50.15 ± 12.27 | 10.47 ± 3.55 | 18.0% | 0.0% | 0.0% |
| SAM | 3.63 ± 3.65 | 84.13 ± 7.94 | 0.13 | 0.82 | 9.09 ± 7.86 | 11.36 ± 7.19 | 68.66 ± 11.36 | 6.35 ± 2.74 | 46.0% | 4.0% | 0.0% |
| OQO | 4.39 ± 4.56 | 79.44 ± 10.31 | 0.15 | 0.78 | 6.28 ± 5.35 | 9.53 ± 4.81 | 59.78 ± 15.10 | 8.06 ± 3.25 | 41.0% | 8.0% | 3.0% |
| Avg. | 3.95 ± 4.61 | 81.84 ± 10.45 | 0.14 | 0.81 | 8.37 ± 8.01 | 10.59 ± 7.47 | 64.00 ± 14.43 | 7.13 ± 3.49 | 47.4% | 4.9% | 1.4% |

### A.11.6 PALLATOM-LIGAND (W/ SA) + MPNN

| | Ca-RMSD | protein-pLDDT | Diversity | Novelty | ligand-Dcenter | ligand-RMSD | ligand-pLDDT | ipAE | Success Rate | | |
| | | | | | | | | | Fold | Pocket | Pose |
|---|---|---|---|---|---|---|---|---|---|---|---|
| FMN | 1.38 ± 1.63 | 89.93 ± 3.82 | 0.13 | 0.81 | 3.21 ± 4.16 | 5.51 ± 4.30 | 72.61 ± 9.21 | 4.40 ± 1.39 | 92.0% | 17.0% | 10.0% |
| DOG | 1.24 ± 0.92 | 89.82 ± 3.61 | 0.10 | 0.82 | 4.12 ± 6.38 | 7.26 ± 5.88 | 71.94 ± 7.33 | 4.45 ± 1.21 | 85.0% | 9.0% | 1.0% |
| LDP | 1.26 ± 1.50 | 94.46 ± 4.53 | 0.11 | 0.85 | 4.60 ± 5.62 | 6.15 ± 5.08 | 80.02 ± 7.47 | 2.84 ± 1.35 | 90.0% | 38.0% | 9.0% |
| SRO | 1.14 ± 0.93 | 93.03 ± 3.98 | 0.09 | 0.84 | 5.88 ± 6.80 | 7.45 ± 6.18 | 77.19 ± 6.74 | 3.26 ± 1.10 | 91.0% | 26.0% | 6.0% |
| IAI | 1.61 ± 2.67 | 89.30 ± 8.12 | 0.15 | 0.80 | 4.49 ± 4.87 | 7.45 ± 5.03 | 69.35 ± 12.59 | 4.92 ± 2.12 | 87.0% | 17.0% | 8.0% |
| FAD | 1.80 ± 1.89 | 85.84 ± 6.39 | 0.24 | 0.78 | 4.81 ± 4.89 | 8.38 ± 5.22 | 65.06 ± 11.12 | 5.82 ± 1.94 | 78.0% | 6.0% | 2.0% |
| SAM | 1.38 ± 1.79 | 90.58 ± 5.28 | 0.13 | 0.82 | 6.82 ± 7.03 | 9.25 ± 6.51 | 77.25 ± 8.17 | 4.15 ± 1.83 | 87.0% | 22.0% | 7.0% |
| OQO | 1.12 ± 0.75 | 90.35 ± 3.89 | 0.15 | 0.78 | 2.60 ± 2.14 | 6.49 ± 3.32 | 68.89 ± 12.12 | 4.97 ± 1.36 | 94.0% | 14.0% | 8.0% |
| Avg. | 1.36 ± 1.63 | 90.41 ± 5.67 | 0.14 | 0.81 | 4.56 ± 5.56 | 7.04 ± 5.37 | 72.78 ± 10.65 | 4.35 ± 1.80 | 88.0% | 18.6% | 6.4% |

### A.12 INFERENCE LATENCY AND SAMPLING EFFICIENCY

In this section, we evaluate the computational cost from two perspectives: the raw inference latency per sample and the total computational budget required to achieve successful designs (sampling efficiency).

**Inference Latency.** First, we compare the inference speed for generating a single protein-ligand complex. The sampling protocol for each method follows the configuration described in Appendix A.9. All benchmarks were performed on a single NVIDIA RTX 5880 GPU with a batch size of 1. As shown in Table 3, our method incurs a marginal increase in latency (135s) compared to RFdiffusion2 (118s).

**Sampling Efficiency.** While diffusion models are inherently iterative, practical usability should be evaluated based on *Sampling Efficiency*—defined as the total computational time required to yield a

| Method | RFdiffusionAA | RFdiffusion2 | Ours |
|---|---|---|---|
| Inference Time (s/sample) | 247s | 118s | 135s |

Table 3: Comparison of inference time per sample (batch size = 1).

fixed number of successful designs. Although our per-sample latency is slightly higher, our method significantly reduces the total computational cost due to a superior design success rate.

We quantify this efficiency by calculating the total time $T_{\text{total}}$ needed to obtain 100 successful candidates:

$$T_{\text{total}} = \frac{100}{\text{Success Rate}} \times t_{\text{inference}} \tag{9}$$

Based on the success rates reported in the main text (3.5% for RFdiffusion2 and 6.4% for Ours):

- **RFdiffusion2:** To obtain 100 successes, it requires $\approx 2,857$ attempts.

$$2,857 \text{ attempts} \times 118 \text{ s/sample} \approx \mathbf{337,126} \text{ s}$$

- **Ours:** To obtain 100 successes, it requires $\approx 1,562$ attempts.

$$1,562 \text{ attempts} \times 135 \text{ s/sample} \approx \mathbf{210,870} \text{ s}$$

Consequently, despite a 14% increase in per-step latency, our model reduces the total computational time by approximately **37%** compared to RFdiffusion2. This demonstrates that for practical design campaigns, our method is the more computationally efficient choice.

## A.13 GENERALIZATION ANALYSIS: RARE AND UNSEEN LIGANDS

A critical challenge in protein-ligand design is the scarcity of training data and the long-tailed distribution of ligand types in the PDB. In this section, we evaluate the generalization capability of Pallatom-Ligand from two perspectives: few-shot learning on rare ligands and zero-shot learning on completely novel small molecules.

### A.13.1 GENERALIZATION TO RARE LIGANDS (FEW-SHOT REGIME)

The training dataset exhibits a highly imbalanced distribution. To assess whether our model merely memorizes abundant protein-ligand pairs or learns generalizable physical patterns, we analyzed the occurrence frequency of the 8 ligands used in our main benchmark (Figure 4 in the main text).

As shown in Table 4, the distribution is extremely skewed. Common cofactors like FAD and FMN appear hundreds to thousands of times, whereas ligands such as DOG, IAI, and OQO are extremely rare. Notably, IAI and OQO appear only once in the entire training set. Despite this scarcity, the results presented in the main text demonstrate that Pallatom-Ligand achieves high design success rates on these rare targets, comparable to its performance on data-rich ligands. This suggests strong few-shot generalization capabilities enabled by our dual-sampling strategy.

| Ligand | Count | Ligand | Count |
|---|---|---|---|
| FAD | 1826 | SRO | 23 |
| FMN | 1569 | LDP | 13 |
| SAM | 599 | DOG | 4 |
| | | IAI | 1 |
| | | OQO | 1 |

Table 4: **Frequency of benchmark ligands in the training dataset** The distribution highlights the few-shot nature of the task for ligands like IAI and OQO.

A.13.2 GENERALIZATION TO NOVEL LIGANDS (ZERO-SHOT REGIME)

To rigorously test the scalability of our method to unseen chemical structures, we conducted additional experiments on four targets that were completely absent from our training set. These ligands represent small-molecule ligands distinct from standard PDB entries:

- **Janelia Fluor Dyes (JF585, JF635, JF711):** Janelia Fluor dyes are synthetic chemical compounds. Their chemical structures have been optimized for fluorescence brightness, photostability and cell permeability.
  - **JF585:** An orange fluorescent dye featuring a **xanthene scaffold** modified with 3,3-difluoroazetidines, representing a distinct structural motif compared to natural metabolites.
  - **JF635:** A far-red dye built on a **silicon-rhodamine (Si-rhodamine)** scaffold. The incorporation of a silicon heteroatom introduces unique steric and electronic properties.
  - **JF711:** A near-infrared dye characterized by a **phosphine oxide-containing rhodamine** scaffold. Its bulky structure poses a significant challenge for binding pocket design.
- **Deschloroclozapine (DCZ):** Deschloroclozapine (DCZ), a metabolite of Clozapine, is a highly potent small-molecule ligand widely used in chemogenetics.

We applied the same evaluation protocol as in the main benchmark. Table 5 compares the Ligand-Pose Success Rate of our method against state-of-the-art baselines.

| Method | JF585 | JF635 | JF711 | DCZ |
|---|---|---|---|---|
| RFdiffusionAA (mpnn1) | 0.0% | 0.0% | 0.0% | 0.5% |
| RFdiffusion2 (mpnn1) | 10.2% | 11.1% | 4.5% | 2.0% |
| Ours (w/out SA) | 5.0% | 0.0% | 1.0% | 0.5% |
| Ours (w/out SA) (mpnn1) | 11.1% | 7.0% | 4.0% | 1.0% |
| Ours (w/ SA) | 3.0% | 2.0% | 4.0% | 2.0% |
| **Ours (w/ SA) (mpnn1)** | **22.4%** | **16.0%** | **17.0%** | **12.0%** |

Table 5: **Ligand-Pose Success Rate on unseen (zero-shot) targets** RFdiffusionAA and RFdiffusion2 serve as baselines. Our method (especially with MPNN sequence redesign) significantly outperforms baselines on these novel chemotypes.

As observed, baseline methods like RFdiffusionAA struggle with these unseen synthetic scaffolds, yielding near-zero success rates. In contrast, our full pipeline (Ours w/ SA + mpnn1) achieves the highest success rates across all four targets. This demonstrates robust zero-shot generalization, confirming that Pallatom-Ligand learns fundamental atomic-level interaction patterns rather than overfitting to specific ligand types seen during training.

A.14 TRAINING OBJECTIVE

Our diffusion framework is primarily based on the Elucidating Diffusion Model (EDM) (Karras et al., 2022). Let $x \in \mathbb{R}^{(N_a) \times 3}$ denote the atomic coordinates of a protein-ligand complex, where $N_a = 14L + l$, $L$ is the number of protein residues, and $l$ is the number of ligand atoms. Furthermore, let $\sigma$ be the standard deviation of the noise, and let $D_{b,ij}(x)$ represent the binned pairwise distance between the central atoms of tokens $i$ and $j$.

Our composite training objective is defined as:

$$L_{\text{obj}} = L_{\text{diff}} + 0.1 L_{\text{seq}} + 0.1 L_{\text{smooth\_lddt}} + 0.03 L_{\text{distogram}} \tag{10}$$

where the individual loss terms are defined as follows:

$$L_{\text{diff}} = \frac{\sigma^2 + \sigma_{\text{data}}^2}{\sigma \cdot \sigma_{\text{data}}} \sum_{i=1}^{N_a} w_i \left\| x_i^{\text{denoised}} - x_i^{\text{gt}} \right\|^2 \tag{11}$$

$$L_{\text{seq}} = \mathbf{1}(\sigma < 0.1) \cdot \text{CrossEntropy}(\text{aatype}^{\text{pred}}, \text{aatype}^{\text{gt}}) \tag{12}$$

$$L_{\text{distogram}} = \text{CrossEntropy}(D_{b,ij}^{\text{pred}}(x), D_{b,ij}^{\text{gt}}(x)) \tag{13}$$

and $L_{\text{smooth\_lddt}}$ is the smooth-LDDT loss as described in AlphaFold3 (Abramson et al., 2024).

The atom-specific weights $w_i$ in the diffusion loss are set as follows: $w_i = 10$ for ligand atoms, $w_i = 3$ for protein atoms within 4 Å of any ligand atom, and $w_i = 1$ for all other protein atoms. This weighting scheme prioritizes the accurate placement of the ligand and its binding pocket.

### A.15 TRAINING DETAILS

The model was trained for 70,000 steps using the AdamW optimizer. We set the learning rate to $1 \times 10^{-3}$, betas to (0.9, 0.95), and weight decay to 0.001. The gradient norm was clipped at 10. Training was conducted on 14 NVIDIA H20 GPUs with a per-GPU batch size of 54 and 2 gradient accumulation steps, resulting in an effective batch size of 108. To balance performance and efficiency, we utilized a mixed-precision strategy: all token-level operations, except for the $QK^T$ einsum, were performed in bfloat16, while all atom-level operations were performed in float32. An Exponential Moving Average (EMA) of the model parameters was maintained throughout training with a decay rate of 0.999.

### A.16 ABLATIONS

In this section, we perform ablation studies to validate the design choices of our model architecture and training strategy.

#### A.16.1 TRANSFORMER ARCHITECTURE

As detailed in Section 3.2, our proposed model employs a "triple-attention" transformer architecture that integrates token-, atom-, and pair-level streams. To elucidate the individual contributions of these components and verify their necessity, we conducted ablation studies on a scaled-down version of the model.

We compared the **Baseline** (consistent with the proposed full architecture) against three variants: (1) **w/o Token Attn**, where the token-level attention is removed; (2) **w/o Atom Attn**, where the atom-level attention is removed; and (3) **w/o Triangle Attn**, where the triangle attention mechanism for pair updates is omitted.

The training dynamics are illustrated in Figure 7, and the quantitative evaluation of sample quality is presented in Table 6. For evaluation, we generated 50 structures for each target in the benchmark with a fixed sequence length of 192. We evaluated the *raw* generated sequences directly without applying MPNN refinement to assess the intrinsic generative capability of the architecture.

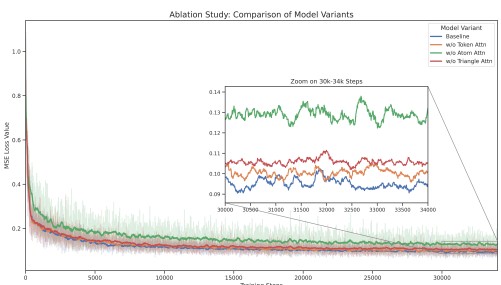

| Model | Ca-RMSD (Å) | Lig-RMSD (Å) |
|---|---|---|
| **Baseline** | **6.78** | **12.00** |
| w/o Token Attn | 7.87 | 14.35 |
| w/o Atom Attn | 9.19 | 13.35 |
| w/o Triangle Attn | 9.91 | 13.69 |

Table 6: **Ablation study on architecture components** Metrics are evaluated on the raw outputs from each model.

Figure 7: **Training dynamics.** The full architecture (Baseline) exhibits faster and more stable convergence compared to ablated variants.

As shown in Table 6, the Baseline architecture yields superior sample quality compared to all ablated variants. These results confirm that the synergistic integration of all three streams is essential for the model's performance.

### A.16.2 Dual-Objective Training Strategy

As discussed in Section 3.4, protein-ligand dataset exhibits a severe distributional dichotomy: clustering by global fold biases the model against diverse ligand interactions, while clustering solely by local interfaces risks structural collapse. To investigate the necessity of our dual-objective formulation, we compared our balanced approach (1:1 ratio) against two extreme sampling regimes:

- **Fold-Centric (Mode 1 only):** Training exclusively with structure-based clustering.
- **Interaction-Centric (Mode 2 only):** Training exclusively with ligand-based clustering.

Crucially, we evaluated performance separately on "Abundant" targets (e.g., FMN, FAD) and "Rare" targets (e.g., DOG, IAI) to highlight generalization capabilities across different data densities. The results are summarized in Table 7.

| Strategy | Ca-RMSD (Å) | | Ligand-RMSD (Å) | |
|---|---|---|---|---|
| | Abundant | Rare | Abundant | Rare |
| Mode 1 only (Fold-Centric) | 5.67 | 5.16 | 13.24 | 14.50 |
| Mode 2 only (Inter.-Centric) | 8.23 | 8.04 | 12.35 | 12.91 |
| **Baseline Dual (1:1)** | 6.79 | 6.85 | 12.93 | 10.59 |

Table 7: **Impact of Training Sampling Strategy.** We report Ca-RMSD (backbone quality) and Ligand-RMSD (binding pose quality) on abundant and rare target subsets. The balanced strategy offers the best trade-off, crucial for generalization to rare ligands.

The results clearly demonstrate the distinct roles of each sampling mode and the synergy of the dual-objective framework:

1. **Limitations of Fold-Centric Training:** While Mode 1 achieves the lowest Ca-RMSD, ensuring high backbone quality, it generalizes poorly to ligand binding on Rare targets (Lig-RMSD degrades to 14.50 Å). This confirms that relying solely on global fold clustering exacerbates data imbalance, causing the model to neglect rare interaction patterns.

2. **Limitations of Interaction-Centric Training:** Conversely, focusing exclusively on binding interfaces (Mode 2) leads to a deterioration in backbone geometry, evidenced by increased Ca-RMSD ($>8$ Å) across both groups. This lack of a stable global fold prevents the formation of a coherent binding pocket, which indirectly compromises interface stability (Lig-RMSD worse than Baseline).

3. **Effectiveness of Balanced Sampling:** The proposed 1:1 ratio effectively acts as a synergistic compromise. It maintains reasonable structural integrity (comparable Ca-RMSD to Mode 1) while significantly narrowing the performance gap on rare targets (Lig-RMSD 10.59 Å v.s 14.50 Å). This validates that the co-existence of both sampling modes is a decisive factor for generalizable design.

### A.17 Impact of Data Imbalance on Model Collapse

Data imbalance is a critical challenge that impairs model generalization and can lead to model collapse. This issue manifests on two primary levels within protein-ligand structural datasets. Firstly, while the protein universe encompasses a vast and diverse folding space, its representation in existing data is highly non-uniform. As illustrated in Figure 6(a), the size distribution of structural clusters exhibits a pronounced long-tail pattern: a small number of extensively studied superfamilies (e.g., kinases, immunoglobulins) account for a disproportionately large fraction of the entries. In contrast, thousands of other clusters that constitute the bulk of structural diversity contain only a handful of members. This structural-level skewness causes the model to be excessively trained on the features of common folds while severely under-exposed to the vast number of rarer ones. Secondly, a similar imbalance exists within the chemical space of ligands and their associated binding patterns. Figure 6(b) demonstrates that the most frequently observed Chemical Component IDs (CCDs) are predominantly common metal ions (e.g., MG, ZN), biological cofactors (e.g., HEM,

ATP), or crystallization artifacts (e.g., GOL). In stark contrast, a vast number of ligands with specific biological activities or those representing potential drug candidates are sparsely represented (Figure 9). This dual imbalance—spanning both structural and chemical space—forces the model to learn statistical artifacts from high-frequency samples rather than the generalizable physicochemical principles of interaction. Consequently, when confronted with novel targets or ligands from the long tail of the distribution, the model's predictive power degrades sharply, leading to the phenomenon of model collapse.

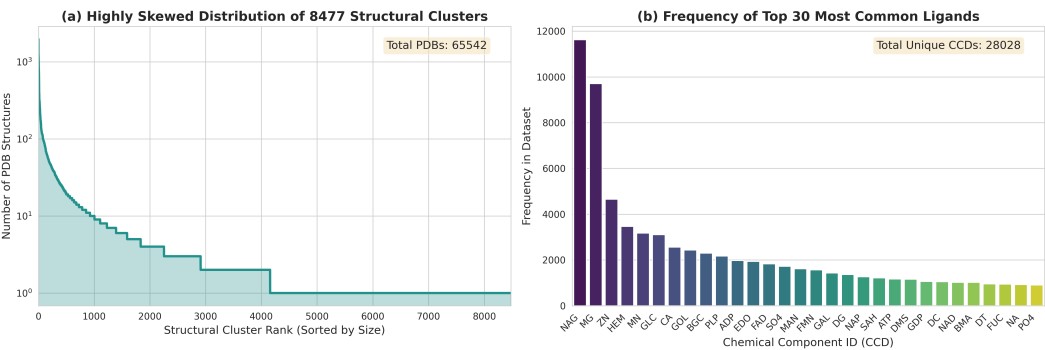

Figure 8: **Imbalance in the Protein-Ligand Structural Dataset** This figure illustrates the severe skewness of the data from two perspectives. (a) The size distribution of protein structural clusters. The x-axis represents structural clusters ranked by size in descending order, and the y-axis (log scale) shows the number of PDB structures within each cluster. The distribution exhibits a characteristic long tail, where a few very large clusters dominate the dataset, while the vast majority of clusters contain very few members. (b) The frequency distribution of the top 30 most common Chemical Component IDs (CCDs). This distribution is also highly imbalanced, with common ions, cofactors, and crystallization agents (e.g., NAG, MG, ZN) occurring far more frequently than other functional molecules. This dual imbalance at both the structural and ligand levels is a key factor that can bias model training towards high-frequency samples and potentially lead to model collapse.

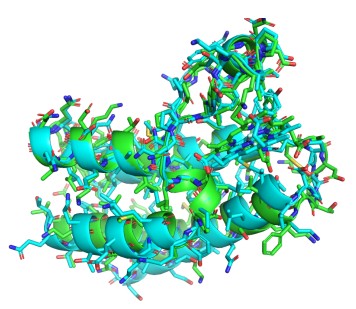

Figure 10: Model collapse leading to low-diversity backbone generation.

In our initial experiments, we observed that training the model with a protein-ligand combination weighting scheme caused it to collapse, resulting in the generation of protein backbones with very low diversity, as shown in Figure 10. We hypothesize this is because diffusion models have a tendency to learn the mean of the data distribution. Compared to the global architecture of the entire protein, the local geometry of the ligand and the protein-ligand interface represents a much higher-frequency signal, causing the model to overfit to this dominant feature.

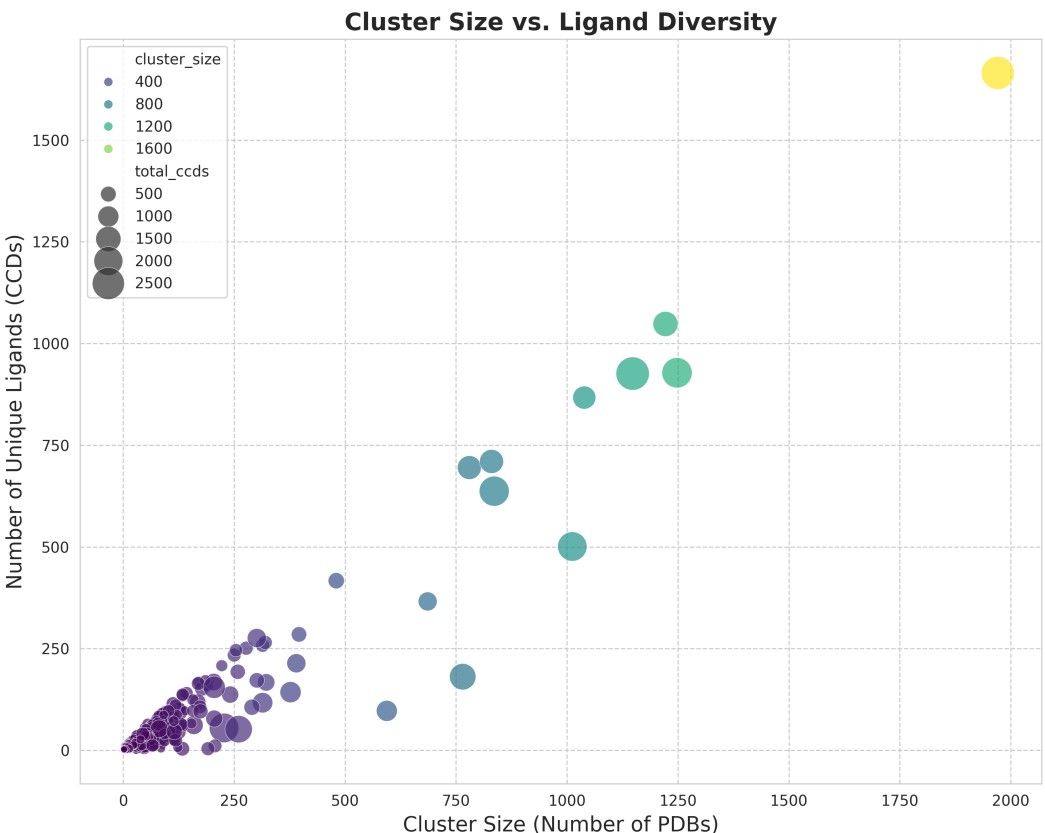

Figure 9: **Correlation between Structural Cluster Size and Ligand Diversity** This scatter plot investigates the relationship between the representation of protein families and their associated chemical space. The x-axis represents the size of each structural cluster (i.e., the number of PDBs), while the y-axis indicates the number of unique ligands (CCDs) found within that cluster. A general positive correlation is observed, suggesting that more extensively studied protein families tend to have a greater number of known ligands. However, the wide dispersion of data points reveals a deeper imbalance in "binding patterns": some large clusters exhibit relatively low ligand diversity (lower-right region), whereas some medium-sized clusters are hotspots of high chemical diversity. This uneven exploration of chemical space across protein families poses a significant challenge for learning generalizable protein-ligand interaction principles.

