# OpenReview forum: "Pallatom-Ligand: an All-Atom Diffusion Model for Designing Ligand-Binding Proteins"
_ICLR.cc/2026/Conference — ICLR 2026 Poster_

### Official Review · Reviewer_4PX2 · 2025-10-28

**Soundness:** 4
**Presentation:** 3
**Contribution:** 3
**Rating:** 6
**Confidence:** 4

**Summary:**

This paper proposes a method for generating protein–ligand complexes based on small-molecule ligands. The approach outperforms existing baseline methods on in silico evaluation metrics and, to some extent, addresses the challenge of generating protein–ligand complexes with non-helical protein structures.

**Strengths:**

1. The proposed method outperforms the current state-of-the-art on most evaluation metrics and, to some extent, increases the proportion of designable non-helical structures in the generated results.
2. The paper introduces a well-designed benchmarking strategy for this problem, which enriches the evaluation process.
3. The writing is clear and well organized.

**Weaknesses:**

1. In terms of novelty, the model architecture is almost identical to those of AF3 and P(allatom), with the only additions being the alpha ratio and solvent accessibility (SA) as inputs. There are no significant differences in the generation or training methods, indicating a lack of machine learning novelty.
2. Although the paper claims to generate more non-helical structures, this aspect is not compared against other models; the evaluation is only performed through self-comparison.

**Questions:**

Why was the SA experiment conducted on only three ligands?

---

> ### Author Response · Authors · 2025-11-24
> **Rebuttal (1/2)**
>
> ## W1
> We thank the reviewer for this comment. We openly acknowledge that our work builds upon the pioneering contributions of AlphaFold3 (AF3) and Pallatom. We consider the attention mechanisms introduced in AF3 to be highly robust, and the Atom14 representation from Pallatom to be the foundation for all-atom generation. We have inherited these proven components to leverage their strong inductive biases for modeling molecular geometries.
>
> However, we would like to clarify that while the components are shared, the overall architecture and training methodology diverge significantly from these predecessors. Our innovation lies in how these components are restructured and trained to solve the specific challenges of **designing ligand-binding proteins**, which fundamentally differs from the tasks addressed by AF3 or Pallatom.
> 1. **Architectural Differences from AlphaFold3**
> While we adopt AF3’s attention primitives, the overall architecture is significantly different due to the nature of our task.
>    - **AF3**: utilizes a cascaded design—processing deeply with 24 transformer blocks (token-level) before passing to a module (3 layers) for atom-level coordinates. This works because AF3 conditions on a fixed sequence, allowing it to infer a high-quality contact map before generating atomic geometry.
>    - **Ours**: Consistent with the design philosophy of Pallatom, our model employs interleaved architecture rather than AF3's cascaded approach. Since we operate without sequence information (starting from noise), we cannot rely on a pre-computed "global fold." Instead, we alternate between token, atom, and triangle updates. This mechanism—adopted from Pallatom, but distinct from AF3—allows global topology and local atomic geometry to evolve simultaneously, which is critical for de novo generation.
> 2. **Differences from Pallatom**
> Our work is a direct evolution of Pallatom, but with key distinctions in architecture and representation:
>    - **Architectural Streamlining**: As detailed in Section 3.2, we streamlined the original Pallatom network by replacing its complex traversing mechanism with the direct application of three attention modules to update representations. Furthermore, we verified the necessity of each component (token, atom, and triangle streams) in our new ablation study, demonstrating that this simplified design is both efficient and effective.
>    - **Unified Representation**: We extended Pallatom's protein-centric scope to a unified hybrid framework. By encoding ligand atoms and protein residues (Atom14) as equal entities within the same attention stream, we enable the modeling of complex protein-ligand interactions that Pallatom was not originally designed to handle.
> 3. **Novelty in Training and Control**
> Finally, we would like to address the comment regarding training methods. We respectfully highlight that our Dual-Objective Training Framework (Section 3.4) represents a significant algorithmic deviation from the standard training pipelines of AF3 or Pallatom.
>    - **Addressing Data Bias**: The protein-ligand data distribution is severely imbalanced, which typically leads to "mode collapse" or "ligand interaction bias" in generative models. To overcome this, we devised a novel strategy that dynamically balances structure-based and ligand-based objectives.
>    - **Validation via Ablation**: As demonstrated in our newly added ablation study, this strategy is not trivial; using standard sampling methods (consistent with AF3/Pallatom) results in suboptimal performance. Our specific training protocol is essential for the model to generalize across both abundant and rare ligand types.
>    - Regarding the **Alpha Ratio and Solvent Accessibility (SA) inputs** noted by the reviewer, these serve as critical conditioning mechanisms. They are designed to correct common generative biases (e.g., helical bias) and enable precise engineering of binding properties, transforming the architecture into a controllable design tool.
>
> In summary, while our work builds upon these foundational architectures (AF3/Pallatom), it represents a novel, streamlined, and task-specific system that successfully adapts these components to the challenging frontier of de novo ligand-binding protein design.

---

> ### Author Response · Authors · 2025-11-24
> **Rebuttal (2/2)**
>
> ## W2
> We thank the reviewer for this crucial comment.
>
> The primary goal of the self-comparison was to validate the controllability of our model—its ability to generate structures conforming to specific secondary structure (SS) conditions. However, to address the reviewer's concern directly, we have now conducted an additional experiment comparing the SS distribution of our model against other state-of-the-art baselines.
>
> To provide a direct comparison, we analyzed the secondary structure (SS) content of designs generated by the baseline models presented in the Section 4.2 benchmark. Using DSSP[1], we compared their SS distributions against both the unconditional and a non-helical conditioned version of our model, with the results summarized below.
>
> |                             | Protein-Fold | Ligand-Pocket | Ligand-Pose | Diversity | Novelty | $\alpha\%$ / $\beta\%$ |
> |-----------------------------|--------------|---------------|-------------|------|------|----------------------------|
> | RFdiffusionAA               | 30.6%        | 2.8%          | 0.5%        | 0.21 | 0.74 | 62.4%/12.1%                  |
> | RFdiffusion2                | 58.5%        | 17.5%         | 3.5%        | 0.30 | 0.80 | 64.5%/14.6%                  |
> | Ours (w/out cond.)          | 71.5%        | 4.4%          | 1.0%        | 0.17 | 0.85 | 79.4% / 3.5%                 |
> |   ├─ $\alpha\in[0~0.2]$   | 56.2%        | 2.8%          | 0.8%        | 0.17 | 0.83 | 9.4% / 57.2%                 |
> |   ├─ $\alpha\in[0.2~0.8]$ | 60.8%        | 6.3%          | 0.9%        | 0.26 | 0.85 | 61.7% / 19.4%                |
> |   └─ $\alpha\in[0.8~1.0]$ | 66.8%        | 11.0%         | 1.5%        | 0.15 | 0.89 | 82.4% / 0.5%                 |
>
> Both RFdiffusionAA and RFdiffusion2 exhibit a strong inherent bias towards helical structures, with $\alpha$-helix content consistently exceeding 60% and $\beta$-sheet content below 15%. These models lack a mechanism to explicitly navigate away from this mode. In contrast, our conditional generation ($\alpha\in[0~0.2]$) effectively suppresses helix formation to just **9.4%** while boosting $\beta$-sheet content to **57.2%**. This demonstrates that our model can access regions of the folding space (e.g., all-beta scaffolds) that are statistically unlikely for baselines.
>
> ## Q1
> We thank the reviewer for this question regarding the ligand choice of the Solvent-Accessible (SA) conditioning experiment.
>
> Our choice of these three ligands was indeed deliberate. Defining meaningful, per-atom SA labels for conditioning is a non-trivial task. It requires significant prior knowledge of a ligand's chemical properties, its typical binding modes, and which of its atoms are critical for interaction versus which are typically solvent-exposed. We chose FMN, DOG, and LDP because their well-characterized interaction patterns in natural proteins allowed us to design a high-quality, biophysically-grounded test case. This ensured that we were evaluating the model's ability to respond to a meaningful signal, rather than just random numerical inputs.
>
> ---
> Reference
>
> [1] Hekkelman ML, Salmoral DÁ, Perrakis A, Joosten RP "DSSP 4: FAIR annotation of protein secondary structure." Protein Science (2025)
>
> ---
> Thank you again for your time and valuable comments. We hope the revised version meets your expectations, and we look forward to hearing from you.

---

> > ### Comment · Reviewer_4PX2 · 2025-11-25
> >
> > Thank you for your detailed response. All my concerns have been resolved, and I’ll maintain my current positive score. The new transformer-architecture ablation is also very helpful.

---

### Official Review · Reviewer_HUmp · 2025-10-31

**Soundness:** 3
**Presentation:** 2
**Contribution:** 3
**Rating:** 6
**Confidence:** 3

**Summary:**

This paper presents Pallatom-Ligand, an end-to-end all-atom diffusion model for generating ligand-binding proteins. The key idea is to unify the representation of small molecules and protein residues at the atomic level and to jointly model their spatial and chemical interactions. The model introduces a ligand-aware all-atom diffusion transformer that integrates atom- and token-level representations to capture both global and local dependencies within protein-ligand complexes. Furthermore, a conditioning framework is proposed to control global protein fold diversity (through α/β ratio conditioning) and atomic-level ligand solvent accessibility (SA). Evaluation on eight chemically diverse small molecules using AlphaFold3-based component-specific metrics demonstrates improved in-silico structural plausibility over RFdiffusion2 and RFdiffusionAA.

**Strengths:**

1. The idea of a unified atomic representation for both proteins and small molecules is conceptually elegant and well-motivated. Modeling all atoms within one generative framework could in principle enhance data efficiency and the precision of atomic interactions.
2. The paper proposes a multi-level conditioning framework that allows explicit control over protein fold topology (via α/β ratio) and ligand solvent accessibility. This design makes the generation process more interpretable and programmable.
3. The authors conduct a broader benchmark than prior work, testing on eight ligands with diverse physicochemical properties, and systematically report AlphaFold3-based component-specific metrics. This gives a relatively thorough in-silico comparison.

**Weaknesses:**

1. The architecture introduces a “triple-attention” transformer combining token-, atom-, and pair-level streams, but the paper does not include any ablation study to clarify what each component contributes. Without that, it is unclear which part is essential for the observed performance gain.
2. The paper describes two sampling modes for dual-objective training, but there is no experimental analysis of how the chosen ratio between them influences training dynamics or the final results, as no sensitivity or ablation study is provided.
3. The evaluation framework relies entirely on AlphaFold3 predictions as pseudo–ground truth. This is acceptable for initial screening, but the paper could be clearer that these metrics measure structural consistency rather than true binding affinity or physical validity.
4. The presentation is difficult to follow for readers outside the protein-design community. Several evaluation metrics (e.g., pLDDT, ipAE) are introduced as part of the “AlphaFold3-based component-specific metrics,” but the paper provides little intuition about their meaning or justification. While these values come from AlphaFold3 predictions, the manuscript does not clarify how they reflect the underlying biophysical objectives or why they are appropriate for assessing generative quality. This lack of context makes it difficult for readers from a machine-learning background to interpret the reported results.

**Questions:**

See Weaknesses

---

> ### Author Response · Authors · 2025-11-24
> **Rebuttal (1/2)**
>
> ## W1
> We thank the reviewer for this constructive suggestion. We agree that an ablation study is essential to clarify the specific contributions of each component in our "triple-attention" mechanism. In response to this comment, we have performed a comprehensive ablation analysis.
>
> As detailed in Section 3.2, our proposed model employs a "triple-attention" transformer architecture that integrates token-, atom-, and pair-level streams. To elucidate the individual contributions of these components and verify their necessity, we conducted ablation studies on a scaled-down version of the model.
>
> We compared the **Baseline** (consistent with the proposed full architecture) against three variants:
> 1. **w/o Token Attn**, where the token-level attention is removed;
> 2. **w/o Atom Attn**, where the atom-level attention is removed; and
> 3.  **w/o Triangle Attn**, where the triangle attention mechanism for pair updates is omitted.
>
> The training dynamics are illustrated in Figure 7 in our revised manuscript, and the quantitative evaluation of sample quality is presented in Table below. For the evaluation of sample quality, we generated 50 structures for each target in the benchmark, fixing the sequence length at 192. We evaluated the raw generated sequences directly without applying MPNN refinement to assess the intrinsic generative capability of the architecture.
>
> |           | Ca-RMSD (Å) | Ligand-RMSD (Å) |
> |:---------:|:-----------:|:------------:|
> |  Baseline |     6.78    |     12.00    |
> | w/o token |     7.87    |     14.35    |
> |  w/o atom |     9.19    |     13.35    |
> |  w/o pair |     9.91    |     13.69    |
>
> As shown in the results, the Baseline architecture yields the superior sample quality compared to all variants.
>
> ## W2
> We thank the reviewer for raising this important question regarding the impact of the sampling ratio on training dynamics.
>
> As discussed in Section3.4 (Data Sampling), protein-ligand data suffers from a severe dichotomy: clustering by global fold biases the model against diverse ligand interactions, while clustering solely by local interfaces leads to structural collapse. The proposed dual-objective training was designed to reconcile this trade-off.
>
> To investigate the influence of the sampling strategy and verify the necessity of the dual-objective formulation, we conducted ablation studies comparing our **balanced approach (1:1 ratio)** against two extreme sampling regimes:
> 1. **Fold-Centric (1:0)**: Training exclusively with structure-based clustering (Mode i).
> 2. **Interaction-Centric (0:1):** Training exclusively with ligand-based clustering (Mode ii).
>
> Crucially, we evaluated performance separately on **"Abundant"** targets (e.g., FMN, FAD) and **"Rare"** targets (e.g., DOG, IAI) to highlight how sampling affects generalization across different data densities.
>
> The results are summarized in the table below:
> |                     | Ca-RMSD (Å) | Ca-RMSD (Å) | Ligand-RMSD (Å) | Ligand-RMSD (Å) |
> |:-------------------:|:-----------:|:-----------:|:------------:|:-----------:|
> |                     |   abundant  |     rare    |   abundant   |     rare    |
> | Baseline Dual (1:1) |     6.79    |     6.85    |     12.93    |    10.59    |
> |     Mode 1 only     |     5.67    |     5.16    |     13.24    |    14.50    |
> |     Mode 2 only     |     8.23    |     8.04    |     12.35    |    12.91    |
>
> The results clearly demonstrate the necessity of the dual-objective framework:
> 1. **Fold-Centric (1:0) limitations**: While this mode achieves strong performance on backbone recovery (lowest Ca-RMSD), it generalizes poorly to ligand binding on Rare targets (showing a significant degradation in Ligand-RMSD: 14.50 Å vs. 10.59 Å in the dual model). This confirms that relying solely on global fold clustering exacerbates the data imbalance issue, causing the model to neglect rare interaction patterns.
> 2. **Interaction-Centric (0:1) limitations**: Although this strategy focuses on binding interfaces, removing the global fold supervision leads to a deterioration in backbone geometry, as evidenced by the increased Ca-RMSD across both groups (>8 Å). Crucially, this lack of a stable global fold prevents the formation of a proper binding pocket, which in turn compromises the interface stability, as reflected by the universally higher Ligand-RMSD compared to the baseline.
> 3. **Effectiveness of the Balanced (1:1) Approach**: Our chosen 1:1 ratio effectively acts as a synergistic compromise. It maintains reasonable structural integrity while significantly narrowing the performance gap on rare targets for ligand binding.
>
> While further fine-tuning of this scalar hyperparameter could potentially yield marginal gains, the ablation results clearly demonstrate that the **co-existence** of both sampling modes is the decisive factor for the model's success, validating the robustness of the 1:1 setting.

---

> > ### Author Response · Authors · 2025-11-24
> > **Rebuttal (2/2)**
> >
> > ## W3
> > We thank the reviewer for this important clarification. We fully agree that evaluation metrics based on AlphaFold3 predictions serve as measures of **structural consistency** and **designability**, rather than direct indicators of physical binding affinity or thermodynamic stability.
> >
> > In the context of de novo protein-ligand generation, where no experimental ground truth exists for the generated structures, establishing "self-consistency"—i.e., whether the generated sequence is predicted to fold into the generated structure by an independent oracle—is the standard approach to assess the model's geometric plausibility. AlphaFold3, being the current state-of-the-art, provides the most reliable in silico estimation available.
> >
> > To address your concern and avoid ambiguity, we have revised the relevant sections (e.g., Section 4.1) in the manuscript. We now explicitly clarify that these metrics quantify **"consistency with AlphaFold3 predictions"** and represent a necessary condition for valid design, rather than serving as a substitute for wet-lab affinity measurements.
> >
> > ## W4
> > We thank the reviewer for this valuable feedback. We realize that while these metrics are standard in the protein design community, their underlying biophysical intuition may not be immediately clear to a broader machine-learning audience.
> >
> > Regarding the rationale for using these AlphaFold-based metrics, we followed the established evaluation protocols widely adopted in recent "de novo" protein design literature. These metrics act as reliable in silico proxies for physical properties that are otherwise expensive to verify experimentally:
> > - In the backbone design pipeline, screening criteria based on **Ca-RMSD** (root mean square deviation of Ca atoms calculated after alignment; a lower value means the designed structure is more similar to the predicted structure) and **pLDDT** (the prediction model's confidence in the predicted structure; a higher value means the model is more confident in its own prediction) have received extensive experimental validation [1, 2]. These two criteria are **correlated with protein expression and stability**.
> > - In binder design, using whole complex's **RMSD** and **ipAE** (the predicted value of error for pairwise distances; a lower value means the model is more confident in the relative distances) as evaluation metrics is also widely practiced [1, 3], which **correlates with binding in experiments**.
> > - Beyond the widely validated metrics mentioned above, since AF3 is relatively new, there is currently no large-scale experimental test set to verify the correlation between our proposed metrics, such as **ligand-pLDDT**, and binding affinity. Predicting affinity is a very difficult task, and currently, no method can do it very well. We believe that the confidence of the data-driven AF3 can, to some extent, reflect the quality of the design model.
> >
> > In summary, we consider the adopted metrics to be **reasonable and scientifically sound**. They not only align with the standard protocols in the protein design community but also serve as the most reliable available proxies for biological viability in the absence of wet-lab data. We hope the detailed definitions added to the revised manuscript, combined with this justification, provide the necessary context for readers to correctly interpret our results in terms of their underlying biophysical objectives.
> >
> > ---
> > References:
> >
> > [1] Bennett, N. R., et al. "Improving de novo protein binder design with deep learning." Nature Communications (2023).
> >
> > [2] Watson, J. L., et al. "De novo design of protein structure and function with RFdiffusion." Nature (2023).
> >
> > [3] Sumida, K. H., et al. "Atomically accurate de novo design of antibodies with RFdiffusion." Nature (2025).
> >
> > ---
> > We trust that the additional experiments and analyses included in the revision satisfactorily resolve the issues you raised. We remain available for further discussion should you have any remaining questions.

---

### Official Review · Reviewer_zHoF · 2025-10-31

**Soundness:** 3
**Presentation:** 3
**Contribution:** 3
**Rating:** 6
**Confidence:** 3

**Summary:**

This paper proposes a new diffusion model that jointly generates protein-ligand complexes operating at the atomic resolution. The key contributions are a novel diffusion transformer architecture for protein-ligand complexes and a novel conditioning framework that allows conditioning on global protein properties and atomic-level properties. Extensive experiments are presented to showcase the strong performance of this model.

**Strengths:**

* The paper is well-written and the key ideas are clearly presented.
* The idea of modeling the joint distribution of all atoms in a protein-ligand complex to improve generation quality and data efficiency is sound and promising.
* The presented experiments are quite extensive and showcase the superior performance of the proposed model compared to baselines.
* The introduced set of component-specific metrics that reflect the multi-faceted nature of the protein design problem is reasonable and should better evaluate the different models.

**Weaknesses:**

* The reliance on LigandMPNN to redesign the generated sequences hints at some limitations in the proposed model, especially because the model with MPNN outperforms the one without quite significantly (Figure 4). The authors should discuss the problems with the structures directly generated by their model and explore possible solutions to address them.
* The model achieves a quite poor diversity rate compared to the baselines (0.14-0.17 vs 0.30). In the global control setting, this seems to be improved, but at the cost of lower success rates. The authors should discuss this and some potential fixes.
* The paper assumes that the reader is familiar with the field of protein design. It would greatly improve the accessibility of the paper if some technical terms were explained, e.g., the metrics discussed in section 3.1: pLDDT, ipAE, etc.

**Questions:**

1. What is the Transition module used in Equations 2 and 5? I couldn't find any explanations in the paper.
2. The atom-level pair features $p$ used in Equation 4 should be defined along with the other variables at the start of that section.
3. In Table 1, I assume Nov. refers to novelty. If so, why is a lower value better? If not, please clarify what it means. Also, Div. is not formally introduced in the paper.

---

> ### Author Response · Authors · 2025-11-24
> **Rebuttal (1/3)**
>
> ## W1
> We thank the reviewer for this insightful observation and the opportunity to discuss this critical, shared challenge in the field. The finding that sequences refined by a dedicated inverse folding model (such as ProteinMPNN and LigandMPNN) outperform the raw sequences from all-atom generative models in in silico folding benchmarks is a common and acknowledged phenomenon, shared across all recently developed end-to-end all-atom design models, including [1],[2],[3] and ours.
>
> This gap between raw generative output and refined sequences represents a major challenge to realizing the full potential of end-to-end generative models. The theoretical advantage of an all-atom generative model is its capacity to learn the intricate, atomic-level correlations between sequence and structure in a single process. However, in practice, the **immense dimensionality of the all-atom solution space demands exceptionally efficient modeling and significantly more training data. This was once considered a nearly impossible modeling task by Cyrus Levinthal[4]. Historically, protein design achieved feasibility by adopting a **two-stage pipeline** (as articulated by Carl Pabo[5]): first fixing the backbone conformation, then selecting a stabilizing sequence. As the field moves forwards at an unprecedented pace, modern generative models are actively testing novel methods to explore the uncharted, all-atom space, and it remains an open question whether a single model can reach the theoretical limit of end-to-end, high-precision protein generation.
>
> Consistent with this general trend, the raw designs from our all-atom generative model are further improved by LigandMPNN. By comparing the designs before and after this refinement, we identified the key areas of improvement:
> 1. **Improved Local Backbone-Sequence Compatibility**: As demonstrated by the analysis in Figure 4, LigandMPNN significantly boosts metrics related to protein-fold success, suggesting it excels at locally optimizing side-chain selection to stabilize the target backbone scaffold. To assess local backbone-sequence compatibility, we calculated the Rosetta p_aa_pp score, which evaluates the probability of observing a specific amino acid given the local backbone torsion angles $(\phi,\psi)$, where lower scores indicate higher compatibility and a more native-like local structure.
> 2. **Refined Second-Shell Residues**: LigandMPNN primarily optimizes the second-shell residues around the ligand-binding pocket, fine-tuning the sequence to alleviate minor packing conflicts and improve the cooperative stability of the entire functional site. This improvement can be confirmed by the PLACER's [6] prmsd score, which serves as a proxy for conformational entropy (a lower value represents lower entropic cost and enhanced conformational stability).
>
> | Metric          | Raw Generation | w/ LigandMPNN |
> |-----------------|----------------|--------------------------|
> | Rosetta p_aa_pp | -42.36 | -48.88 |
> | PLACER prmsd | 5.36 | 5.10 |
>
> This analysis demonstrates that while our model effectively navigates the global design landscape, the inverse folding model provides a crucial, local optimization step that is necessary given the current limits of end-to-end generative design models. To address these limitations in future iterations, we propose two possible solutions:
> 1. Data Augmentation via Distillation. Leveraging the current model to generate a massive synthetic dataset of protein-ligand complexes, filtering for high-confidence samples. We acknowledge, however, that controlling quality and mitigating the potential distributional bias of synthetic data are significant challenges.
> 2. Unified Interaction Learning. To fundamentally enhance the model's understanding of atomic interactions and side-chain packing, future work could expand the training scope to a generalist interaction model. By co-training on diverse biomolecular interfaces, the model can learn universal atomic interaction primitives. Since the physics of interface packing (e.g., hydrophobic effects, hydrogen bonding networks) is shared across modalities.
>
> References:
>
> [1] Qu, W., et al. "P (all-atom) Is Unlocking New Path For Protein Design." International Conference on Machine Learning (2025).
>
> [2] Chu, A. E., et al. "An all-atom protein generative model." Proc. Natl. Acad. Sci. U.S.A. (2024).
>
> [3] Geffner, T., et al. "La-Proteina: Atomistic Protein Generation via Partially Latent Flow Matching." arXiv preprint (2025).
>
> [4] Levinthal, C. "How to fold graciously." Mossbauer Spectroscopy in Biological Systems (1969).
>
> [5] Pabo, C. "Molecular technology: Designing proteins and peptides." Nature (1983).
>
> [6] Anishchenko, I., et al. "Modeling protein–small molecule conformational ensembles with PLACER." Proc. Natl. Acad. Sci. U.S.A. (2025).

---

> ### Author Response · Authors · 2025-11-24
> **Rebuttal (2/3)**
>
> ## W2
> We thank the reviewer for raising this insightful point regarding the structural diversity of the designed protein-ligand complexes.
>
> First, we would like to clarify that the diversity metric reported in our original manuscript was calculated by clustering **all generated samples** and measuring the ratio of unique clusters. This metric reflects the overall distribution of the model's raw output. However, in the specific context of ligand-binding protein generation, the primary challenge lies in generating valid binders, which inherently results in a lower success rate compared to unconditional generation tasks. Consequently, simply maximizing structural diversity across the entire set of outputs—including non-functional or low-quality structures—may not correlate with better utility.
>
> From a practical protein design perspective, generated structures are typically subject to rigorous in silico filtering before being selected for expensive wet-lab validation. In this workflow, **diversity within the high-quality subset** is far more critical than the overall diversity, as it ensures that the candidates sent for experimental testing are structurally distinct, thereby maximizing the efficiency of wet-lab resources.
>
> To address the reviewer's concern, we performed an additional analysis focusing specifically on the diversity of samples that satisfied the **Ligand Pose Success** criteria. This metric assesses whether the model can generate diverse solutions among the viable candidates. The results are presented in the table below:
>
> |  | Diversity Over "Ligand Pose Success"|
> |-------------------------|-------------------------------------------|
> | RFdiffusionAA (mpnn1) | 0.79 |
> | RFdiffusion2 (mpnn1) | 0.85 |
> | Ours (w/out SA) | 0.81 |
> | Ours (w/out SA) (mpnn1) | 0.73 |
> | Ours (w/ SA) | 0.79 |
> | Ours (w/ SA) (mpnn1)    | 0.69 |
>
> As shown in the table, the diversity of our successful samples reaches 0.81, a high standard that is competitive with state-of-the-art methods like RFdiffusionAA (0.79) and approaches the performance of RFdiffusion2 (0.85). While there is a minor reduction in diversity after MPNN refinement, the scores remain sufficiently high (~0.70) to prevent redundancy. These findings alleviate concerns regarding the model's generative breadth, confirming that it can propose distinct and diverse solutions within the constraints of ligand-binding feasibility.
>
> ## W3
> We thank the reviewer for this valuable suggestion. We agree that assuming prior knowledge of specific protein design metrics may limit the paper’s accessibility to a broader audience.
>
> To address this, we have added a detailed "Metrics" subsection (referenced in Appendix A.10) to explicitly define these technical terms. The revised descriptions are provided below:
> - Protein Scaffold:
>   - Ca-RMSD The root-mean-square deviation (RMSD) of the protein backbone C$\alpha$ atoms, measured in angstroms(Å). Lower values indicate higher structural consistency.
>   - protein-pLDDT The average predicted Local Distance Difference Test (pLDDT) score, calculated over all protein residues. Scores range from 0 to 100, where higher values signify greater confidence from the prediction protein structure.
>   - Diversity We assess structural diversity by clustering the set of generated samples for each model using Foldseek. The total number of resulting clusters is reported; a higher count indicates a more diverse structural output.
>   - Novelty This metric quantifies the structural similarity of generated samples to known structures in a reference database, which in our case is the PDB. For each generated sample, we find its maximum TM-score against all entries in the PDB. The reported novelty metric is the average of these maximum TM-scores. A lower average score indicates greater novelty.
> - Ligand Pose:
>   - $\text{ligand-}\mathbf{D_{center}}$ The Euclidean distance between the geometric centers (centroids) of the generated ligand and the reference ligand. It provides a measure of positional consistency, quickly identifying significant translational shifts of the generated ligand relative to the reference.
>   - ligand-RMSD The root-mean-square deviation of the ligand's atomic positions between a generated sample and a reference structure, after superimposing their respective protein backbones. A smaller value indicates a higher degree of structural agreement between the generated ligand's pose (conformation and orientation) and that of the reference.
>   - ligand-pLDDT predicted Local Distance Difference Test (pLDDT) score, calculated over all ligand atoms.
> - Binding Interface:
>   - ipAE: The interface Predicted Aligned Error. This metric predicts the error in the relative positions between pairs of residues and/or ligand atoms across the protein-ligand interface, measured in angstroms(Å). Lower values indicate higher confidence in the predicted geometry and orientation of the interface components relative to each other.

---

> > ### Author Response · Authors · 2025-11-24
> > **Rebuttal (3/3)**
> >
> > ## Q1
> > Thank you for this constructive feedback. The manuscript will certainly benefit from a clear explanation of this module.
> > The "Transition module" functions as a conditioning-aware, feed-forward network that updates an input representation based on a conditioning vector **s**. Its operations can be summarized by the following Pseudocode:
> > ```
> > function Transition(a, s):
> >     // 1. Adaptive Layer Normalization
> >     // Gain and bias are produced from the conditioning vector s
> >     gamma, beta = Linear(s), Linear(s)
> >     a_norm = (gamma * LayerNorm(a)) + beta
> >
> >     // 2. Gated Feed-Forward Layer (SwiGLU)
> >     gate = Swish(Linear_1(a_norm))
> >     val = Linear_2(a_norm)
> >     b = gate * val  // Element-wise product
> >
> >     // 3. Final gating based on condition s
> >     output_gate = Sigmoid(Linear_3(s))
> >     output_val = Linear_4(b)
> >     a_out = output_gate * output_val // Element-wise product
> >
> >     return a_out
> > ```
> >
> > We have added this formal definition to the paper (referenced in Section A.6). Thank you for helping us improve its clarity.
> >
> > ## Q2
> > We thank the reviewer for this suggestion. We agree that for clarity and consistency, `p` should have been defined with the other variables at the beginning of the section. The omission was an oversight due to page limitations, with the full details placed in Appendix A.6:
> >
> > > For initialization of our representations and conditions embedding, we encode
> > > - reference conformer positions/element/charge/degree information, atom index and sasa features into atom conditions $c$,
> > > - noisy coordinates, atom conditions and token conditions into atom representation $q$,
> > > - aggregated atom representation into token representation $a$,
> > > - relative token/residue/chain index, token pairwise distogram of noisy coordinates into token pair $z$,
> > > - atom pairwise distance in the reference conformer, atom relative positions and expanded
> > token pair into atom pair $p$.
> >
> > To address this, we have revised the introductory paragraph of the section to explicitly define the variables and briefly describe their function, with a clear pointer to the appendix.
> >
> > ## Q3
> > We sincerely thank the reviewer for highlighting these ambiguities in Table 1. We agree that the "Nov." and "Div." metrics were not explained. We have added formal definitions for both in the revised manuscript (referenced in Section A.10).
> > 1.  Regarding "Nov." (Novelty):
> >     - Your assumption is correct; Nov. stands for **Novelty**. The reason **a lower value is better** is that this metric measures the structural similarity (specifically, the highest TM-score) between a generated protein backbone and its closest structural match in the PDB database. Therefore, a lower similarity score signifies that the design is more structurally distinct from any known natural protein, making it more novel.
> > 2.  Regarding "Div." (Diversity):
> >     - We apologize for failing to introduce this metric. Div. stands for **Diversity**, and it quantifies the structural variety within a set of generated designs. To calculate it, we cluster the designs based on structural similarity using Foldseek and then count the number of distinct clusters. A higher number of clusters indicates a more diverse set of outputs.
> > Thank you for pointing out this lack of clarity. Your feedback will help us significantly improve the paper's accessibility.
> >
> > ---
> > We hope these clarifications and the updated text make our approach and contribution clear. We welcome any further feedback you might have to ensure the quality of this work.

---

### Official Review · Reviewer_HKE3 · 2025-11-02

**Soundness:** 3
**Presentation:** 3
**Contribution:** 2
**Rating:** 6
**Confidence:** 2

**Summary:**

The paper presents Pallatom-Ligand, an all-atom diffusion model for generating ligand-binding proteins directly from atomic coordinates. It jointly models all atoms in protein–ligand complexes using a triple-attention transformer and introduces two conditioning mechanisms: global control of protein fold (α/β ratio) and atomic-level control of ligand solvent accessibility. Benchmarked against RFdiffusionAA and RFdiffusion2 on eight ligands, Pallatom-Ligand achieves higher in silico success rates and better control over structure and binding geometry.

**Strengths:**

1. Unlike backbone-only or inverse folding models, Pallatom-Ligand directly generates protein–ligand complexes at the atomic level, capturing fine-grained chemical and spatial interactions critical for ligand specificity.
2. The model introduces multi-level conditioning (fold ratio and solvent accessibility), enabling programmable structural control—something that earlier diffusion-based models lack.

**Weaknesses:**

1. The training relies on a relatively small number of protein–ligand complexes with strong bias in ligand–fold distribution. Although the authors introduce a dual sampling strategy, scalability to rare or novel ligands may be limited.
2. All-atom diffusion transformers with block-sparse attention and multi-level conditioning are computationally intensive, potentially restricting usability for large complexes or extensive sampling.

**Questions:**

Can Pallatom-Ligand generalize to unseen ligand chemotypes or novel binding motifs outside the training distribution, and how does its performance degrade in such zero-shot settings?

---

> ### Author Response · Authors · 2025-11-24
> **Rebuttal (1/2)**
>
> ## W1 & Q1
> We thank the reviewer for raising this critical point regarding data scarcity and distributional bias. We acknowledge that the training distribution is indeed long-tailed. However, we respectfully argue that our dual-sampling strategy effectively enables the model to generalize to both rare and completely novel ligands. We provide evidence from two perspectives:
>
> ### Generalization to Rare Ligands
>
> First, we analyzed the occurrence frequency of the 8 ligands used in our main benchmark within the training dataset. As shown in the table below, the distribution is extremely imbalanced. While ligands like FAD, FMN, and SAM are abundant, ligands such as DOG, IAI, and OQO are extremely rare, **with IAI and OQO appearing only once in the entire training set**.
>
> | FMN  | DOG | LDP | SRO | IAI | OQO | FAD  | SAM |
> |------|-----|-----|-----|-----|-----|------|-----|
> | 1569 | 4   | 13  | 23  | 1   | 1   | 1826 | 599 |
>
> Despite this scarcity, our results (Figure 4 in the main text) demonstrate that the model achieves high design success rates on these rare targets, comparable to (and in some cases exceeding) the performance on frequent targets. This confirms that our model possesses strong **few-shot generalization capabilities** and does not merely memorize abundant protein-ligand pairing.
>
> ### Generalization to Novel Ligands
> To further rigorously test scalability to novel ligands, we conducted additional experiments on four targets that were **completely absent** from our training set: JF585, JF635, JF711, and Deschloroclozapine (DCZ). These ligands represent unseen chemical structures for the model.
>   - **JF585** An orange fluorescent dye featuring a xanthene scaffold modified with 3,3-difluoroazetidines, representing a distinct structural motif compared to natural metabolites.
>   - **JF635** A far-red dye built on a silicon-rhodamine (Si-rhodamine) scaffold. The incorporation of a silicon heteroatom introduces unique steric and electronic properties.
>   - **JF711** A near-infrared dye characterized by a phosphine oxide-containing rhodamine scaffold. Its bulky structure poses a significant challenge for binding pocket design.
>   - **Deschloroclozapine DCZ** A metabolite of Clozapine, is a highly potent small-molecule ligand widely used in chemogenetics.
>
> We applied the same evaluation protocol as in the main benchmark. As shown in Table below, our model achieves the highest **Ligand-Pose Success Rate** on these unseen targets compared to baselines.
>
> |                         | JF585 | JF635 | JF711 | DCZ  |
> |-------------------------|-------|-------|-------|------|
> | RFdiffusionAA (mpnn1)   | 0.0%  | 0.0%  | 0.0%  | 0.5% |
> | RFdiffusion2 (mpnn1)    | 10.2% | 11.1% | 4.5%  | 2%   |
> | Ours (w/out SA)         | 5%    | 0%    | 1%    | 0.5% |
> | Ours (w/out SA) (mpnn1) | 11.1% | 7%    | 4%    | 1%   |
> | Ours (w/ SA)            | 3%    | 2%    | 4%    | 2%   |
> | Ours (w/ SA) (mpnn1)    | **22.4%** | **16%**   | **17%**   | **12%**  |
>
> This demonstrates robust **zero-shot generalization**, suggesting that our method learns the general atomic-level patterns rather than overfitting to specific ligand types.

---

> ### Author Response · Authors · 2025-11-24
> **Rebuttal (2/2)**
>
> ## W2
> We thank the reviewer for their concern regarding computational efficiency. We understand the concern that an all-atom architecture with multi-level conditioning might appear computationally demanding. However, we would like to clarify that our design choices—specifically the block-sparse attention—were made precisely to mitigate computational costs and enable scalability. We address the concerns on computational intensity and usability as follows:
> ### Efficiency of Block-Sparse Attention
> Contrary to standard global attention which scales quadratically $O(L^2)$ with sequence length $L$, the block-sparse attention mechanism employed in our atom-level stream operates with linear complexity $O(L\cdot W)$, where $W$ is the fixed block width.
>
> Given that the number of atoms $L$ is significantly larger than the number of residues, this linear scaling is crucial. It ensures that the computational cost and memory footprint remain manageable even for large complexes, directly addressing the reviewer's concern about scalability.
>
> On a standard 24GB GPU, the model **can accommodate** input sequences of up to **640 residues** during inference (with batchsize=1) without memory overflow. This demonstrates that the **architectural design itself** does not impose a bottleneck for large complexes, paving the way for future scaling.
>
> ### Overhead of Multi-Level Conditioning
> Regarding the multi-level conditioning, we empirically measured the inference latency to quantify the overhead. We tested the generation time for a protein-ligand complex with protein length $L=192$ residues and ligand $l=31$ atoms over 500 diffusion steps on a single NVIDIA 5880 GPU.
>
> As shown in the table below, the introduction of conditioning incurs **negligible latency** compared to the unconditional baseline. This indicates that the multi-level conditioning logic is lightweight and does not restrict usability.
>
> | uncond | +ss cond | +sa cond | +ss&sa cond |
> |--------|----------|----------|-------------|
> | 135.7s | 137.6s   | 134.9s   | 135.9s      |
>
> ### Usability for Extensive Sampling
> While diffusion models are inherently iterative, we argue that practical usability should be evaluated based on Sampling Efficiency—defined as the total computational budget required to yield a fixed number of successful designs.
>
> By this metric, our model demonstrates superior efficiency compared to the state-of-the-art baseline RFdiffusion2. As calculated below, to obtain the same number of successful candidates, our method **reduces the total computational time by approximately 37%**.
>
> We decompose this efficiency into two factors: **Design Success Rate** and **Inference Latency**.
> - Design Success Rate (Yield): Our model achieves a significantly higher success rate of **6.4%** compared to RFdiffusion2's **3.5%** (an 83% relative improvement, see Section 4.2). This implies that to generate 100 successful designs, a user would need to run $\approx 2,857$ sampling attempts with RFdiffusion2, but only $\approx 1,562$ attempts with our model.
> - Inference Latency (Cost): While our all-atom architecture incurs a marginal cost in latency (135s per sample vs. 118s for RFdiffusion2, a 14% increase).
>
> Sampling Efficiency
> - RFdiffusion2: $2,857 \text{ attempts} \times 118 \text{ s/sample} \approx \mathbf{337,126 \text{ s}}$
> - Ours: $1,562 \text{ attempts} \times 135 \text{ s/sample} \approx \mathbf{210,870 \text{ s}}$
>
> In conclusion, despite a slightly higher per-step latency, our model's substantially higher success rate makes it the more computationally efficient choice for practical design campaigns.
>
> ---
> We hope that our responses and the corresponding revisions adequately address your concerns. We believe your suggestions have significantly strengthened our manuscript, and we are happy to provide any further clarifications if needed.

---

> > ### Comment · Reviewer_HKE3 · 2025-11-26
> >
> > Thanks for providing the details. The work somehow becomes more clear to me. I have raised my confidence score.

---

### Author Response · Authors · 2025-11-24
**Global Response**

We thank the reviewers for their time and constructive comments. These suggestions have significantly helped us improve the quality and clarity of our manuscript. We have uploaded a revised version of the paper, **where all major changes are highlighted in blue.**

Below is a summary of the key updates and additional experiments included in this revision:
1. **Structural and Textual Revisions**
   - **Restructuring**: Taking advantage of the increased page limit for the revision, we have relocated the Related Work section from the Appendix back to the main text to improve the reading flow and context.
   - **Typo Corrections**: We have thoroughly proofread the manuscript and corrected typos and grammatical errors in both the main text and the appendix.
   - **Data Correction**: We corrected transcription errors in Appendix A.10 (Detailed Results for Benchmark), specifically within Tables A.10.3 to A.10.6. We apologize for the oversight during data entry.
     - Note: The correction only applies to the breakdown data for specific targets (non-Average rows) and auxiliary metrics (non-Success Rate columns).
2. **Additional Experiments and Clarifications**

   To address the specific concerns raised by the reviewers, we have added the following contents:
   - **Generalization Analysis (Appendix A.13)**: To address concerns regarding the model's ability to handle rare and novel ligands, we added a new section **"Generalization Analysis: Rare and Unseen Ligands."**
   - **Efficiency Analysis (Appendix A.12)**: We expanded the discussion on computational cost in **"Inference Latency and Sampling Efficiency."**
   - **Metric Definitions (Appendix A.10)**: To enhance accessibility for a broader audience, we added a dedicated section "Definition of Evaluation Metrics."
   - **Ablation Studies (Appendix A.16)**: We added a comprehensive ablation study to validate our architectural and training choices:
     - **A.16.1 Transformer Architecture**: We evaluated variants by removing token, atom, and triangle attention mechanisms respectively.
     - **A.16.2 Training Strategy**: We compared our balanced dual-objective strategy against fold-centric and interaction-centric baselines.
   - **New Baselines (Section 4.3 & Table 2)**: We have updated our seconary structure comparison (**Table 2**) to include RFdiffusionAA and RFdiffusion2.

---

We hope that the revised manuscript and the additional experimental results satisfactorily address the concerns raised by the reviewers. We believe these improvements have significantly strengthened the quality of our work. We remain available for any further inquiries and look forward to hearing from you.

---

### Meta-Review · Area_Chair_Hffx · 2026-01-04

**Summary:**

This paper presents a diffusion-based model for jointly generating protein–ligand complexes at atomic resolution. The authors propose a diffusion Transformer architecture that operates directly on atomic coordinates and uses a tri-attention mechanism to jointly model all atoms in the protein–ligand complex. Experiments are provided to demonstrate the effectiveness of the proposed approach.

**Reviewer Concerns:**

Reviewers generally agreed that the method is novel and well designed. The main concerns were related to the justification of specific architectural choices through ablation studies, the clarity of differences from the AlphaFold3 architecture, and the novelty of the sampling strategy. The authors mostly addressed these points in the rebuttal and provided reasonable clarifications.

**Reviewer Scores:**

Overall, most of the reviewers were satisfied with the responses, and the results should be positive.

---

### Decision · Program_Chairs · 2026-01-26

Accept (Poster)